# BacBench: Evaluating Genomic Language Models for Bacteria

## Abstract

Bacteria underpin key processes in health, ecology, and biotechnology, yet machine learning in bacterial genomics lacks systematic, large-scale evaluation resources. Current resources are typically limited to single-species datasets, where the small number of available genomes leaves species-specific models underpowered, underscoring the need for approaches that can generalize across the bacterial tree of life. To address this gap, we present BacBench, the first comprehensive benchmark for bacterial genomics. BacBench consists of 11 datasets across 6 tasks, including a newly generated dataset for operon identification derived from long-read RNA sequencing. BacBench covers gene-, system-, and genome-scale prediction tasks, spanning 67k genomes, 17.6k species and 255M proteins. We analyze the performance of state-of-the-art DNA LMs, protein LMs and bacterial LMs and find that while each approach excels at different scales—the existing models fail to accurately predict the bacterial phenotype at a whole-genome level, hampering the translation to high-impact applications such as antibiotic-resistance and bioproduction. Therefore, highlighting the need to develop methods that reason over the context of the entire genomes, exploiting genomic synteny and transfer across species. We outline the key requirements for such models and release a standardized library for preprocessing, embedding, and evaluation, fostering the development of methods that accurately represent bacterial genomes, and enabling reproducible comparison of diverse approaches under a unified framework. By providing the first comprehensive benchmark dedicated to bacterial genomics, BacBench lays the ground-work for developing machine learning models that truly exploit shared evolutionary patterns and generalize across the bacterial tree of life.

## 1 Introduction

Bacteria drive indispensable processes in medicine, ecology, and biotechnology (de Steenhuijsen Piters et al., 2015; Luo et al., 2024). They produce industrial enzymes and antibiotics (Ariaeenejad et al., 2024; Santos-Júnior et al., 2024), recycle nutrients, and are being engineered for carbon capture and waste remediation (Xu & Jiang, 2024). Unlocking this potential hinges on interpreting bacterial genomes at scale. A machine learning (ML) system that can embed and reason over entire bacterial genomes could predict clinically relevant traits, surface novel enzymes for biomanufacturing, and reveal how genetic variation translates into functional capabilities across species.

Traditionally, ML approaches in bacterial genomics have been species-specific. For example, genome-wide association studies (GWAS) have been successful in identifying genotype–phenotype associations within species, such as antimicrobial resistance or virulence traits (Lees et al., 2016; Power et al., 2017; San et al., 2020). However, species-specific datasets typically include only a few genomes, leaving models statistically underpowered and prone to overfitting given the vast mutation space of bacterial genomes.

Meaningful progress therefore requires models that can share information across species. Such transfer is biologically plausible: every bacterium carries a small core of universal single-copy proteins, and many additional gene families are conserved far beyond the species level (Wang et al., 2022; Lang et al., 2013; Coleman et al., 2021). Leveraging these shared signals allows models to capture the full extent of bacterial diversity, leading to predictions that are both robust and generalizable. Training on genomes from many species, laboratories, and environmental contexts

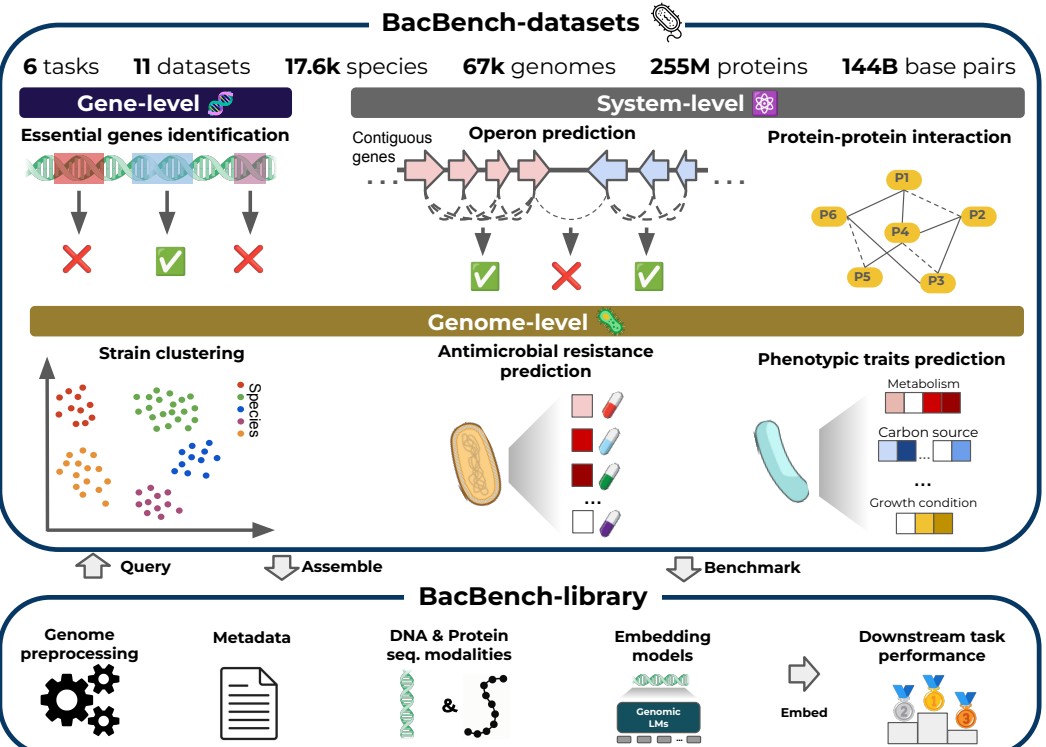

Figure 1: **BacBench overview.** We collected a diverse set of tasks at gene-, system- and genome-scales, spanning 17.6k bacterial species. `BacBench-library` provides support to preprocess and embed the datasets with various models. Finally, we performed systematic benchmarking for each task using diverse genomic LMs.

further guards against dataset-specific artifacts, making the resulting models less prone to sampling bias and more reliable when deployed on previously unseen strains or sequencing pipelines.

Recent breakthroughs in genomic sequence modeling show that the resulting representations can generalize across species and capture evolutionary signals (Nguyen et al., 2024; Dalla-Torre et al., 2024; Zhou et al., 2023; Lin et al., 2022; Elnaggar et al., 2021; Lin et al., 2023; Hayes et al., 2025). However, within the bacterial domain specifically, evaluation has been fragmented—either on narrow, single-task applications like antimicrobial resistance prediction (Wiatrak et al., 2024) or as part of cross-kingdom evaluations that fail to address bacteria-specific challenges (Nguyen et al., 2024). Consequently, the field lacks a dedicated, multi-scale benchmark for bacterial genomics, where genomic and metabolic mechanisms differ substantially from eukaryotes. Thus, leading to the development of models which can accurately model whole bacterial genomes.

Here, we introduce **BacBench**, the first multi-scale, multi-task and multi-species benchmark designed to evaluate ML for bacterial genomics (Fig. 1). BacBench has been collated from a diverse set of 11 datasets organized into 6 tasks spanning multiple biological scales: gene ✂, system ⚛ and genome 🦠. We consider essential gene prediction task at the gene-level, operon and protein–protein interaction prediction tasks at the system-level; and strain clustering, antibiotic resistance, and phenotypic traits prediction tasks at the genome-level. Collected from a diverse set of public resources and newly generated data, these datasets encompass 67k genomes spanning more than 17.6k bacterial species and 255M proteins. Using BacBench, we conduct comprehensive evaluation of distinct approaches to modeling bacterial genomes including DNA LMs, protein LMs (pLMs) and bacterial LMs (bLMs). We find that different modeling approaches excel at different tasks and scales, yet all models achieve low performance on phenotype prediction at a whole-genome level. Our results demonstrate *(i)* the need to develop ML approaches which can accurately model entire bacterial genomes, *(ii)* the benefits of pretraining on bacteria-specific corpora, rather than cross-kingdom ones, thus learning genomic mechanims that are unique to bacteria, and *(iii)* the importance of selecting the right model for the task at hand.

BacBench datasets and an accompanying toolkit for preprocessing and evaluation ensure straight-forward reuse and extension[1]. By generating and integrating these datasets, we introduce the first benchmark suite for bacterial genomics, aiming to catalyze the development of ML methods that can transfer knowledge across species, unlocking new discoveries in bacterial genomics.

## 2 RELATED WORK

Overall, existing resources for evaluating ML in bacterial genomics are limited in several key ways: they often lack functional labels, restrict evaluation to single species, or evaluate only one biological scale. BacBench addresses these limitations by providing a multi-scale and multi-task benchmark across the bacterial tree of life. It is specifically designed to reflect the core challenges in the field, including data sparsity within individual species, the need for cross-species generalization, and the importance of assessing model performance from the level of individual genes to entire genomes.

**Bacterial genomics datasets.** Large-scale sequence repositories such as MGnify (Mitchell et al., 2023), IMG/M (Markowitz et al., 2012), and AllTheBacteria (Blackwell et al., 2024) catalogue millions of bacterial assemblies. These resources are indispensable for comparative genomics, yet they provide limited metadata labels, making them ill-suited for training or benchmarking ML models. Simultaneously, task-specific collections provide information on essential genes and phenotypic traits (Zhang et al., 2004; Madin et al., 2020; Brbić et al., 2016; Weimann et al., 2016; Consortium, 2022)—supply richer annotations but usually cover only a handful of species and a single prediction setting. Diverse Genomic Embedding Benchmark (DGEB) (West-Roberts et al., 2024) offers tasks drawn from all domains of life, yet its bacterial coverage is mostly limited to single-species gene or short-segment datasets and lacks genome-scale evaluations such as broad phenotypic traits inference or strain-level clustering, leaving it unable to assess whether models generalize across thousands of species at multiple scales. BacBench complements these efforts by integrating six heterogeneous tasks, ranging from gene essentiality to genome-wide phenotype prediction - into a unified framework that explicitly tests generalization across 17.6 k bacterial species and three biological scales (Fig. 1).

**Single-species bacterial genomics models.** Despite the existence of large-scale bacterial genomics datasets, the ML applications in bacterial genomics have been mostly confined to single-species and single-task problems. For instance, genome-wide association studies (GWAS) have been successful in identifying genotype–phenotype associations within species, such as antimicrobial resistance or virulence traits (Lees et al., 2016; Power et al., 2017; San et al., 2020). More recent approaches such as unitig-based and deep learning models improved genotype–phenotype mapping by spanning the full pangenome (Lees et al., 2018; 2020) and predicting the effect of mutation based on the genomic context (Wiatrak et al., 2024). While the single-species models often perform well in their domains, they do not extend to other taxa and new genomic variants, and the huge genomic feature space relative to the number of labelled isolates leaves them prone to overfitting.

**Genomic LMs.** **DNA LMs** learn DNA sequence representations and have been shown to accurately represent long sequences (Zhou et al., 2023; Dalla-Torre et al., 2024; Jiang et al., 2023; Mourad, 2025; Nguyen et al., 2024; 2023), but are usually evaluated on human regulatory tasks, where transcriptional mechanisms and epigenomic regulation differ substantially from bacteria (Casadesús & Low, 2006). Moreover, even the DNA LMs with very large context window cannot span entire medium-sized bacterial genomes (Brixi et al., 2025). **Protein LMs** (pLMs) learn representations that correlate with structure, stability, and function (Elnaggar et al., 2021; Lin et al., 2022; 2023; Hayes et al., 2025). As bacterial genomes consist largely of coding sequence and possess simpler regulatory architectures than eukaryotes, pLMs can capture a substantial fraction of relevant biology. pLMs model proteins in isolation, however, characterizing bacterial genomes requires modeling the contextual interactions between the proteins present in the genome. Finally, we differentiate a third group of genomic LMs - **Bacterial LMs (bLMs)** which are recently proposed genomic LMs that are purposefully built to model bacterial genomes. These include gLM2 (Cornman et al., 2024), a mixed-modality genomic LM that represents coding regions as amino acids and intergenic regions as nucleotides to model contiguous genome context, and Bacformer, a genome-level contextual protein LM that treats each bacterial genome as an ordered sequence of proteins, refining each protein vector in the presence of

---

[1]https://anonymous.4open.science/r/BacBench-B6EF

Table 1: Summary of benchmarked models. "Max ctx." = maximum context length supported at inference; "dim" = dimensionality of the output of the last hidden layer. DNA LMs, pLMs and bLMs are separated by a horizontal line.

| Model | Input | Objective | Tokenisation | Params | dim | Training corpus | Max ctx. |
|---|---|---|---|---|---|---|---|
| Mistral-DNA (Mourad, 2025) | DNA | Autoregressive | Byte-pair | 138M | 768 | Bacteria | 512 |
| DNABERT-2 (Zhou et al., 2023) | DNA | Masked | Byte-pair | 117M | 768 | Multi-kingdom | 512 |
| Nucleotide Transformer (Dalla-Torre et al., 2024) | DNA | Masked | $k$-mer | 250M | 768 | Multi-kingdom | 2,048 |
| ProkBERT (Ligeti et al., 2024) | DNA | Masked | $k$-mer | 27M | 384 | Bacteria | 4,096 |
| Evo (Nguyen et al., 2024) | DNA | Autoregressive | Single nucleotide | 6.5B | 4,096 | Multi-kingdom | 8,192 |
| ESM-2 (Lin et al., 2022) | Single protein seq. | Masked | Single amino acid | 35M | 480 | Multi-kingdom | 1,024 |
| ESM-C (ESM Team, 2024) | Single protein seq. | Masked | Single amino acid | 300M | 960 | Multi-kingdom | 1,024 |
| ProtBERT (Elnaggar et al., 2021) | Single protein seq. | Masked | Single amino acid | 420M | 1,024 | Multi-kingdom | 1,024 |
| gLM2 (Cornman et al., 2024) | Mixed modality (DNA & protein seq.) | Masked | Single nucleotide/amino acid | 650M | 1,280 | Bacteria | 4,096 |
| Bacformer (Wiatrak et al., 2025) | Multiple protein seq. | Masked | Single protein | 27M | 480 | Bacteria | 6,000 |

all other proteins from the same genome, thus, encoding organism-level context. Both methods model the DNA or protein in the context of a bacterial genome and are pretrained on extensive bacterial corpora.

## 3 BACTERIAL GENOME REPRESENTATIONS & BASELINES

We selected a diverse set of five DNA LMs, three pLMs and two bLMs to represent bacterial genomes and evaluate their performance across distinct tasks and scales. These genomic LMs take as input DNA, proteins, or both (gLM2) and can therefore generalize across the bacterial tree of life. Moreover, the suite spans modalities (DNA-only, single-protein, mixed DNA–protein, genome-level protein), objectives (masked vs. autoregressive), and training corpora (bacteria-specific vs. cross-kingdom), enabling a controlled comparison of how context length, modality, and pretraining data shape genome embeddings (Table 1).

**Bacterial genomes representations.** For the gene- and system-level tasks with **DNA LMs** we embed the coding sequence plus upstream promoter of the gene; we split sequences longer than the model limit $L$ into overlapping windows of length $L$ and average their embeddings across all windows. For genome-level tasks, we tile each genome into chunks with overlap, embed each chunk, and average the results to extract a genome embedding (Appendix B). For the **pLMs** we embed each protein present in the bacterial genome independently and average its residue embeddings. To generate genome-scale representations, we calculate the mean of all protein vectors (Appendix B). Finally, for **bLMs** we use contiguous, genome-aware inputs: for **gLM2**, we feed mixed-modality genomic segments that encode coding regions as amino acids and intergenic regions as nucleotides, preserving local genome context across adjacent genes; for **Bacformer**, we represent each genome as an ordered sequence of proteins by obtaining per-protein tokens from its base pLM (ESM-2 35M), and pass these through a transformer to learn contextualized, genome-level protein embeddings.

## 4 PREDICTION TASKS AND EVALUATION RESULTS

In BacBench, we consider six tasks across three scales: *(i)* gene ✎, *(ii)* system ❄, and *(iii)* genome 🌿. At gene-level, we consider the task of gene essentiality. At system-level, we assess operon identification and protein-protein interaction prediction tasks. Finally, at genome-level we evaluate methods on strain clustering, antibiotic resistance and phenotypic traits prediction tasks. We briefly describe each task and include further experimental details in the Appendices A & B.

### 4.1 GENE ESSENTIALITY PREDICTION ✎

Identifying essential genes is crucial for *(i)* defining the minimal set of cellular functions, and *(ii)* prioritizing drug targets. To distinguish essential from non-essential genes, the methods need to generalize to phylogenetically diverse bacteria beyond the species in the training set. For each model we report the performance by *(i)* fitting a linear classifier on top of the frozen gene embeddings, *(ii)* finetuning the model to predict a binary label (i.e. essentiality of input gene; Appendix B).

**Data.** We compile the dataset from the Database of Essential Genes (DEG) (Zhang et al., 2004), a hand-curated resource which aggregates studies published for a broad range of bacteria. After quality-control filtering, the corpus comprises 51 distinct genomes spanning 37 species (Appendix

A). This amounts to $22,486$ essential and $146,922$ non-essential genes. To prevent train-test leakage, we split by genus—placing all genomes from a genus in one split—and evaluate on held-out genera, enforcing generalization to phylogenetically distant strains.

**Metrics.** Gene essentiality prediction is a binary classification task. We evaluate performance using AUROC and AUPRC and report macro-average metrics across test genomes.

**Results.** pLMs and bLMs substantially outperform DNA-based models in terms of both AUROC (Fig. 2) and AUPRC using both linear probing and finetuning (Appendix C), suggesting that essential genes share conserved protein motifs that the protein-based embeddings capture. bLMs perform the best, with gLM2 achieving the best results overall, and Bacformer significantly outperforming its backbone ESM-2 model, showing the benefits of incorporating genome context. When considering DNA LMs only, Evo achieves the best performance, demonstrating the benefits of scaling model and context size.

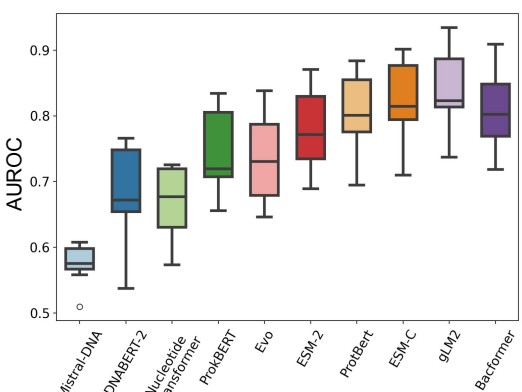

Figure 2: AUROC across genomes on essential gene prediction task using a linear model. The box spans the interquartile range with a line marking the median value.

## 4.2 OPERON IDENTIFICATION

Operons are multi-gene transcriptional units that underpin coordinated gene expression. Accurate operon maps are pivotal for *(i)* constructing gene regulatory networks (Fortino et al., 2014) and *(ii)* refining genome-scale metabolic models for strain engineering (Orth et al., 2010). In this task, the methods must predict whether the two neighbouring genes belong to the same operon, effectively predicting operon boundaries. Because available annotations are scarce we evaluate the task in a zero-shot setting.

**Data.** Due to the lack of experimentally validated operon annotations, we generated the operon labels by performing long-read RNA sequencing on a set of 5 diverse strains, amounting to $3,310$ unique operons (Appendix A).

**Metrics.** In BacBench, operon identification is a zero-shot binary classification task. We use

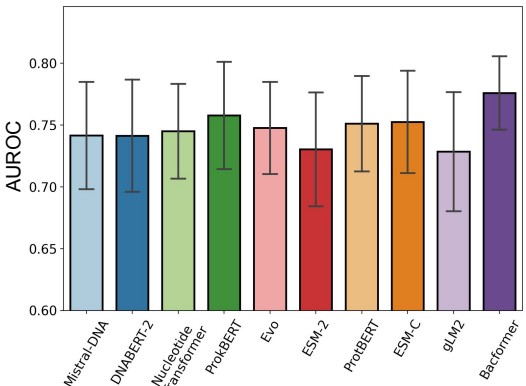

Figure 3: Zero-shot AUROC on operon identification task, the error bars represent standard error across strains.

the cosine similarity value between the two genes combined with the information on the genes' strand as a score indicating whether the genes belong to the same operon (Appendix A). We leverage AUROC and AUPRC for measuring performance. Finally, we report results across distinct strains.

**Results.** All methods except Bacformer attain similar performance (Fig. 3) in terms of both AUROC and AUPRC (Appendix C). We attribute the high performance of the Bacformer due to the *(i)* whole-genomic context of the model and *(ii)* extensive bacterial training corpus. ProkBERT performs 2nd best, showing that *(i)* scaling the model does not necessarily lead to improved results, *(ii)* the importance of a relevant pretraining corpus (ProkBERT was pretrained only on prokaryotes). Notably, both DNA and pLMs perform similarly on the task, indicating that the choice of modality is less important than incorporating genome-level context and domain-matched pretraining. Finally, the low overall performance across models highlights the need for task-specific methods and generating datasets which would allow for finetuning.

## 4.3 Protein-protein interaction prediction ⚛

Mapping protein–protein interactions (PPI) is central to *(i)* reconstructing gene networks (Snider et al., 2015) and *(ii)* prioritizing drug-target combinations (Wilson et al., 2022). Compared to the operon identification task where interacting genes lie next to each other, in the PPI task proteins can be separated by millions of base pairs. In this task each input is a pair of proteins, and the goal is to predict whether two proteins interact. Because interaction data are available at the protein-level, we restrict the benchmark to models which only require protein sequence data, specifically, pLMs and Bacformer. We evaluate two regimes: *(i)* zero-shot: the cosine similarity between the frozen embeddings of the two proteins serves as the interaction score, *(ii)* finetuned: the averaged

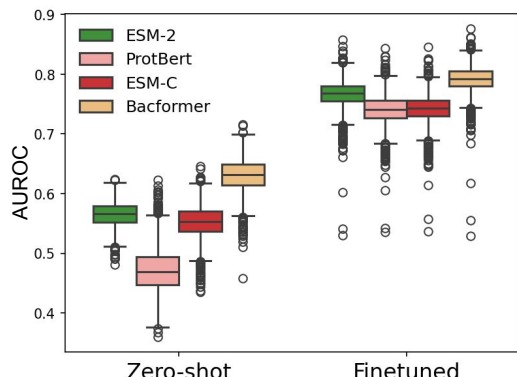

Figure 4: AUROC across genomes on PPI task in the zero-shot and finetuned setting. The box spans the interquartile range with a line marking the median value.

embeddings of the two proteins are passed through a linear classifier trained to output a binary class.

**Data.** We downloaded and processed all $10,533$ bacterial strains available in STRING DB (Szklarczyk et al., 2023) together with associated PPI scores. For every strain we extract the combined interaction score for each protein pair; the median strain contains almost $640,000$ scored pairs. We binarize the labels, resulting in roughly 10% of all interactions being positive (Appendix A).

**Metrics.** PPI is a binary classification task. We report performance with AUROC and AUPRC, macro-averaged over test genomes. Both the positive and negative labels are provided by STRING.

**Results.** In both zero-shot and finetuned setups, Bacformer achieves substantially higher scores than other methods. We attribute this to the rotary positional embeddings (Su et al., 2024), which increase the cosine similarity score between neighbouring genes. This is in line with experimental data, which shows that the neighbouring genes in the bacterial genomes tend to interact with each other (Dandekar et al., 1998). We also notice how the difference in performance between the methods decreases following finetuning, demonstrating how the models can adapt to the task during training. Finally, the performance of ESM-2 compared to ESM-C and ProtBERT shows that scaling up the training data and model size does not benefit PPI prediction in bacteria.

## 4.4 Strain clustering 🦠

Rapid clustering of whole genomes is valuable for placing newly sequenced metagenome assembled genomes (MAGs) into the bacterial tree of life and quality control. Therefore, we propose a metagenomic strain clustering task where the goal is to recover taxonomy using genome only. In this task, we feed every MAG to the model without using any species tokens or other metadata—so evaluation is fully zero-shot. We then evaluate whether models' genome embeddings preserve the taxonomy. A good embedding should cluster genomes from the same species close together and, at broader levels, conserve members of e.g. the same genus or family. We perform the clustering by computing $k$ nearest neighbors across different $k$ and running Leiden clustering (Traag et al., 2019) at various resolutions (Appendix B).

**Data.** In this BacBench task, we draw $6,071$ strains from MGnify (Mitchell et al., 2023), spanning 25 species distributed across 10 genera and 7 families chosen to give a balanced phylogenetic spread. We process DNA and protein inputs in the same way for every model; and average the vector from the final hidden layer over the whole genome to yield one fixed-length embedding per genome.

**Metrics.** We quantify clustering performance with the adjusted Rand index (ARI), normalized mutual information (NMI) and average silhouette width (ASW). Higher is better for all metrics. To obtain cluster assignments for each model we run Leiden clustering across resolutions, retaining the resolution that maximized the mean of the three metrics over the species, genus and family ranks. The ASW is unaffected by taxonomic rank, so we report it once in the *Combined* section.

Table 2: Clustering performance (higher is better) across taxonomic ranks and metrics. Best value for each metric is bolded. DNA LMs, pLMs and bLMs are separated by a horizontal line.

| Method | Species | | Genus | | Family | | Combined | | |
|---|---|---|---|---|---|---|---|---|---|
| | ARI | NMI | ARI | NMI | ARI | NMI | ARI | NMI | ASW |
| **Mistral-DNA** | 89.57 | 94.85 | 69.04 | 86.60 | 48.52 | 78.56 | 69.04 | 86.67 | 39.88 |
| **DNABERT-2** | 97.10 | 98.33 | 64.98 | 85.81 | 43.85 | 76.35 | 68.64 | 86.83 | 63.10 |
| **Nucleotide Transformer** | 98.14 | 98.89 | 64.06 | 85.28 | 43.14 | 75.84 | 68.45 | 86.67 | 39.24 |
| **ProkBERT** | **98.75** | **99.38** | 63.55 | 84.88 | 42.76 | 75.46 | 68.35 | 86.58 | **65.03** |
| **Evo** | 55.56 | 76.98 | 49.28 | 72.42 | 35.53 | 66.58 | 46.79 | 72.00 | 25.33 |
| **ESM-2** | 50.94 | 71.67 | 56.82 | 72.20 | 46.84 | 69.06 | 51.53 | 70.97 | 16.75 |
| **ESM-C** | 72.65 | 87.70 | 79.76 | 89.87 | 60.39 | 83.43 | 70.93 | 87.00 | 29.55 |
| **ProtBERT** | 68.53 | 86.47 | **85.98** | **92.40** | **65.78** | **86.42** | **73.43** | **88.43** | 39.12 |
| **gLM2** | 72.33 | 84.03 | 66.60 | 82.27 | 49.03 | 75.93 | 62.65 | 80.74 | 40.16 |
| **Bacformer** | 79.39 | 90.62 | 77.92 | 90.20 | 54.74 | 80.71 | 70.68 | 87.17 | 30.12 |

**Results.** At the species level, DNA LMs attain the highest scores, indicating that species boundaries can be recovered from sequence alone and consistent with the extra signal carried in non-coding regions. Moving up to genus and family levels, the advantage shifts: pLMs and Bacformer overtake the DNA LMs, suggesting that protein embeddings retain deeper evolutionary relationships more faithfully. Aggregated across all ranks, ProtBERT shows the highest ARI and NMI values, whereas the DNA models have the overall highest ASW. Interestingly, gLM2 which is a mixed modality model combining DNA and protein sequences performs worse than other models. Upon investigation, we believe this is due to the large variance in its embeddings and suggest that a mixed modality method with improved regularization could capture both fine-grained strain identity and higher-order phylogeny in a single embedding space.

## 4.5 ANTIBIOTIC RESISTANCE PREDICTION 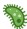

Predicting antibiotic resistance is a task with *(i)* immediate clinical value for guiding antimicrobial drug therapy, *(ii)* monitoring the spread of resistant lineages and *(iii)* prioritizing compounds in drug-discovery pipelines. In this BacBench task, we define two subtasks: (1) given a bacterial genome, predict whether the strain is resistant or susceptible to a specific drug, and (2) given a bacterial genome, estimate its minimum inhibitory concentration (MIC). The first subtask is a binary classification problem (resistant vs susceptible), while the second subtask is a regression problem. Due to the computational complexity of computing genome-level embeddings on over $25k$ genomes (Appendix B), we *(i)* do not include Evo in the analysis due to its size (6.5B parameters), which makes it computationally infeasible to embed the entire corpus in most academic environments (see *Runtime analysis*; Appendix B), *(ii)* perform evaluation by stacking a linear layer on top of the frozen genome representation, and fine-tune a separate linear classifier for each model.

**Data.** We assemble a cross-species panel from the NIH Antimicrobial Susceptibility Test browser (National Center for Biotechnology Information, 2025), covering $25,032$ strains drawn from 38 bacterial species. After quality control and removal of sparsely sampled drugs, the dataset retains 36 antibiotics for binary label and 56 for regression prediction that span diverse classes of antibiotics (Appendix A). For the binary task we use the resistant/susceptible calls and discard any ambiguous entries (Appendix A). For the regression task we extract the raw MIC values, apply a $log1p$ transformation to dampen heavy tails and train a separate linear model for each drug–model pair.

**Metrics.** We report performance in the classification setting with AUROC and AUPRC across drugs. For the MIC regression we compute the *Pearson* correlation coefficient and the coefficient of determination ($R^2$) averaged across antibiotics. We report mean scores across antibiotics and include full per-antibiotic tables in the Appendix C.

**Results.** On both binary and regression setup, bLMs and pLMs tend to outperform DNA LMs. This may be explained by the fact that resistance is usually acquired through the mutation in the coding sequence, with studies showing that 90% of characterized resistance-conferring variants reside in coding regions (Sandgren et al., 2009; Farhat et al., 2019). Finally, the bLM-Bacformer achieves the best results implying the importance of epistatic effects on antibiotic resistance (Trindade et al., 2009) which the model considers by modeling the interactions between all of the proteins present in the genome. Notably, the methods record highly variable performance across antibiotics, underscoring the difficulty of building a single model that generalizes across the resistance mechanisms.

Table 3: Performance (%, higher is better) on the antibiotic-resistance prediction tasks across drugs. Values are mean ± standard deviation across 3 runs; best for each metric is bolded. DNA LMs, pLMs and bLMs are separated by a horizontal line.

| Method | Binary | | Regression | |
|---|---|---|---|---|
| | AUPRC | AUROC | $R^2$ | Pearson |
| **Mistral-DNA** | 49.86 ± 22.73 | 76.05 ± 9.14 | 19.71 ± 17.37 | 40.95 ± 21.51 |
| **DNABERT-2** | 52.35 ± 23.55 | 79.88 ± 7.26 | 23.61 ± 18.33 | 45.89 ± 19.27 |
| **Nucleotide Transformer** | 59.70 ± 22.90 | 84.22 ± 6.47 | 27.74 ± 17.80 | 51.19 ± 16.76 |
| **ProkBERT** | 58.90 ± 23.67 | 83.94 ± 6.39 | 28.00 ± 17.57 | 51.24 ± 16.91 |
| **ESM-2** | 62.04 ± 23.64 | 85.05 ± 6.63 | 31.18 ± 17.39 | 54.60 ± 16.15 |
| **ESM-C** | 63.41 ± 23.43 | 85.68 ± 6.81 | 33.15 ± 17.62 | 56.79 ± 15.46 |
| **ProtBERT** | 61.79 ± 23.19 | 84.51 ± 6.31 | 28.23 ± 18.46 | 51.56 ± 17.61 |
| **gLM2** | 55.80 ± 24.46 | 82.16 ± 6.75 | 26.15 ± 18.13 | 49.41 ± 17.14 |
| **Bacformer** | **67.97** ± 20.73 | **87.61** ± 5.68 | **33.84** ± 18.60 | **57.53** ± 15.43 |

## 4.6 PHENOTYPIC TRAITS PREDICTION

Accurately predicting phenotype from a genomic sequence enables *(i)* inference of the biological or ecological function of bacteria (Feldbauer et al., 2015) and *(ii)* engineering organisms with the exact metabolic traits needed for efficient waste remediation (Rafeeq et al., 2023), accelerating sustainable industrial bioproduction (Lawson et al., 2021). We evaluate whether the models' genome embeddings are predictive of diverse phenotypic traits. Here, given a genome embedding, the task is to predict a trait. Similarly as in the antibiotic resistance prediction task above, we perform linear probing evaluation, training a separate linear classifier per phenotype and exclude the Evo model due to the computational costs (Appendix B).

**Data.** To create the benchmark we collated large trait inventories (Madin et al., 2020; Brbić et al., 2016; Weimann et al., 2016), apply stringent quality filters and discard traits represented by only a handful of isolates (Appendix A). The final corpus covers 139 discrete phenotypes spanning 15, 477 bacterial species, making it the broadest dataset of its kind and challenging the models to generalize well beyond their training clades. We group traits into carbon utilisation, biochemical activity, growth conditions and cellular morphology (Appendix A), which we use later for stratified analysis. As similar genomes often share the same phenotype, we split the data for each phenotype by genus—placing all genomes from a genus in one split—and evaluate on held-out genera, enforcing generalization to phylogenetically distant strains.

**Metrics.** For BacBench, we restrict the phenotypic traits to categorical traits due to their sufficient number of available samples, and evaluate performance using the macro-averaged AUROC and AUPRC over phenotype categories. We report mean scores for every phenotype group and include full per-trait tables in the Appendix C.

**Results.** pLMs and bLMs outperform DNA LMs across phenotype groups and metrics (Fig. 5 & Appendix C). The relative difference is especially large for biochemical activity and carbon utilization traits, likely because these phenotypes hinge on enzyme active-site composition and pathway membership—information that is explicit in amino-acid space and can be better captured

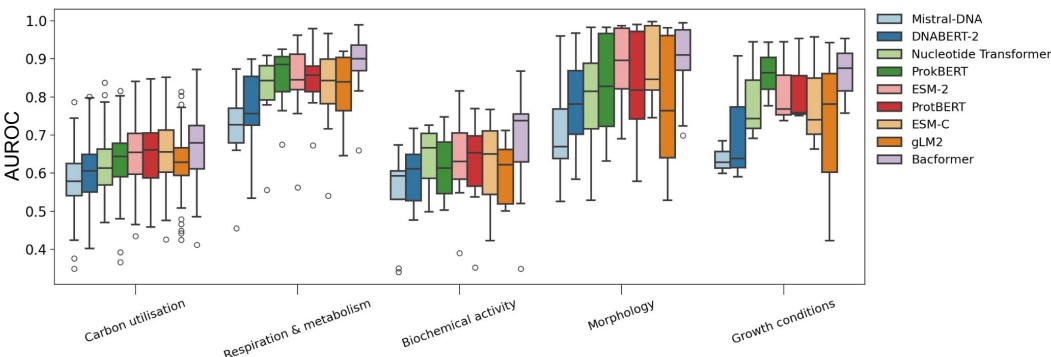

Figure 5: AUROC across diverse phenotypic traits groups and methods. The box spans the inter-quartile range with a line marking the median value.

by pLMs (Teukam et al., 2024; Lawson et al., 2021). The Bacformer bLM attains the highest scores, indicating that whole-genome context helps explain phenotypes arising from coordinated protein function and epistatic interactions (Trindade et al., 2009). In contrast, Mistral-DNA, the only autoregressive model in the task, lags well behind the other models. Overall performance remains moderate—especially for carbon utilization, biochemical activity and growth condition traits—highlighting considerable room for improvement. Progress will likely require more complex contextualized models that incorporate environmental metadata, together with larger, more balanced phenotype datasets.

## 5 DISCUSSION

**Summary.** BacBench addresses the lack of a comprehensive, cross-species evaluation resource for bacterial genomics by providing unified datasets and benchmarks, and by evaluating existing genomic language models across species, tasks, and biological scales. It introduces a newly generated dataset for operon identification—a key problem for refining genome-scale metabolic models—and curates five additional tasks (gene essentiality, protein–protein interaction, strain clustering, antibiotic resistance, and 139 phenotypic traits) into a single framework covering 67k genomes from 17.6k species. Our experiments show that *(i)* existing genomic LMs (DNA LMs, pLMs, and bLMs) capture core taxonomic structure and functional relatedness, providing a strong baseline representation of bacterial genomes, but fall short at accurate genome-to-phenotype prediction, as evidenced by antibiotic resistance and phenotype tasks; *(ii)* models purpose-built for bacteria (gLM2, Bacformer) or trained on bacteria-specific corpora (ProkBERT) tend to outperform broad, cross-kingdom counterparts across most tasks, underscoring the value of domain-matched pretraining and inductive biases; *(iii)* different modeling approaches excel at different problems—DNA LMs capture fine-grained taxonomic signals and do well on operon identification, pLMs better preserve deeper phylogeny and functional similarity for strain clustering, and bLMs like Bacformer perform best on tasks driven by multi-gene interactions (phenotypes, antibiotic resistance). All datasets are accompanied with extensive documentation; the embedding and evaluation library is provided at https://anonymous.4open.science/r/BacBench-B6EF.

**Towards accurate genome-to-phenotype bacterial prediction.** Our results suggest a practical path forward towards building model for genome-to-phenotype mapping in bacteria: *(i)* the model should represent entire genomes end-to-end to capture long-range, cross-protein dependencies (currently only feasible in Bacformer); *(ii)* pretrain on substantially larger, bacteria-focused corpora (we estimate >4M unique strains remain untapped (Mitchell et al., 2023; Markowitz et al., 2012; Blackwell et al., 2024)); *(iii)* integrate DNA and protein modalities—and where available RNA—to couple regulatory and coding signals in a single embedding while remaining computationally efficient; *(iv)* allow inclusion of structured priors (e.g., resistance gene catalogs, operon maps, HGT markers) to improve data efficiency on scarce phenotype labels; *(v)* expand high-quality phenotype supervision (including knock-outs and standardized trait panels) to close the supervision gap limiting genome-to-phenotype learning.

**Limitations & Future Work.** By releasing BacBench we provide a foundation for more expressive, cross-species models of bacterial genomics, yet the present benchmark covers only part of the functional landscape. Other tasksa re not yet included, and certain phenotypes and antibiotic classes remain sparse (Appendix A), underscoring the need for generating new data. We benchmarked a representative set of publicly available models capable of cross-species generalization, and expect the model suite to expand in future iterations. Finally, the current tasks do not include modalities such as transcriptomics and metabolomics due to data sparsity and inconsistent metadata. As community datasets mature, incorporating multi-omics will enable more faithful evaluation of causal, context-dependent genome-to-phenotype mappings.

We anticipate that subsequent iterations will broaden task and model coverage, ultimately enabling contextual, genome-scale representations for bacterial genomes. Community contributions of datasets, models, and evaluation routines are encouraged so that BacBench evolves into a continually updated standard for bacterial ML.

## 6 REPRODUCIBILITY STATEMENT

We release an *anonymous* codebase for preprocessing, embedding, and evaluation at `https://anonymous.4open.science/r/BacBench-B6EF`, together with helper utilities for bacterial genomics to lower the barrier for adding new models and tasks. All datasets in BacBench are fully documented and accompanied by anonymized MLCommons `Croissant` metadata files in the supplementary materials, enabling unambiguous data loading and lineage tracking. The Appendix details quality filtering, preprocessing pipelines, train/validation/test splits, model and training configurations, and hyperparameters; it also specifies random seeds, hardware, and optimization settings. Further details on the exact scripts to reproduce the results can be found in the anonymous repository linked above. These artifacts together provide the necessary pointers—code, data descriptors, and experimental specifications—to reproduce our results and to extend BacBench as an evolving benchmark for the ML community.

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

## A ADDITIONAL DATASETS & TASKS DETAILS

In this section we outline the dataset details, including the overall and per dataset statistics as well as the preprocessing details. For each dataset we performed additional quality checks which we detail below. All of the genomes across datasets contain genome ID, which can be used to identify the genome source. Finally, we provide metadata on genes, proteins and genomes together with the datasets when available, together with extensive documentation.

Supplementary Table 1: Dataset summary for the six benchmark tasks used in this study.

| Task | # Species | # Genomes | # Proteins | # Base pairs |
|---|---|---|---|---|
| Gene essentiality prediction | 37 | 51 | 169 k | 279 MB |
| Operon identification | 5 | 5 | 22 k | 25 MB |
| Protein–protein interaction prediction | 6 956 | 10 533 | 36 M | N/A |
| Strain clustering | 25 | 6 710 | 14 M | 16 GB |
| Phenotypic traits prediction | 15 477 | 24 462 | 100 M | 111 GB |
| Antibiotic resistance prediction | 38 | 26 302 | 105 M | 112 GB |

### A.1 GENE ESSENTIALITY PREDICTION

We downloaded the gene essentiality annotations for bacteria across genomes from the Database of Essential Genes (DEG, http://origin.tubic.org/deg) (Zhang et al., 2004). Using the genome RefSeq ID provided in the database, we downloaded the associated genomes in both DNA and protein sequence modalities from the NCBI GenBank (https://www.ncbi.nlm.nih.gov/genbank/). Across 66 genomes from DEG, there were multiple genomes with more than 98% overlap when it comes to annotations. We therefore removed these genomes, as including it could lead to inflated evaluation metrics, leaving us with 51 genomes across 37 distinct species. For each genome we provide start and end for each gene together with essentiality annotations (*Yes*=essential, *No*=non-essential), verifying the gene locations are correct. We also provide the strand of the gene to allow for the extraction of the region upstream of the gene.

**Split.** We performed a random data split into training, validation, and test sets in a 60 / 20 / 20 % ratio. Additionally, to prevent train-test leakage, we split by genus—placing all genomes from a genus in one split—and evaluate on held-out genera, enforcing generalization to phylogenetically distant strains.AUROC

### A.2 OPERON IDENTIFICATION

Due to the lack or reliable whole-genome operon annotations, we performed long-read RNA sequencing to annotate operons across five distinct strains, processing three independent biological replicates for each strain.

**Bacterial strains and culture conditions.** Five bacterial strains were used in this experiment: *Staphylococcus aureus RN450* (*S. aureus* RN450), *Mycobacterium abscessus* ATCC 19977 (*M. ab*), ΔleuD ΔpanCD *Mycobacterium tuberculosis* H37Rv 102J23 (ΔleuD ΔpanCD *M. tb*), *Pseudomonas aeruginosa* PAO1, and *Escherichia coli DH5α*. Each strain was cultured in triplicate under nutrient-rich conditions until mid-exponential phase ($OD_{600} = 0.4$–$0.6$) was reached.

**RNA isolation.** Cells were harvested by centrifugation and processed for total RNA extraction using the MasterPure Complete DNA and RNA Purification Kit (Lucigen) with strain-specific modifications: For *S. aureus* RN450, cell pellets were pre-treated with lysostaphin (Tris buffer, pH 8.0) at 37 °C

for 30 min to aid lysis. For the mycobacterial strains (*M. ab* and ΔleuD ΔpanCD *M. tb*), cells were mechanically disrupted by bead beating in lysis buffer, followed by extraction with the standard kit protocol. Isolated RNA was treated twice with TURBO DNase (Invitrogen) to remove residual genomic DNA and purified using RNA Clean & Concentrator columns (Zymo Research). RNA integrity was assessed on an Agilent TapeStation RNA ScreenTape system, and concentrations were measured with a Qubit fluorometer (Invitrogen).

**Library preparation and sequencing.** For each replicate, 1,000 ng of total RNA underwent rRNA depletion with riboPOOLs (siTOOLs Biotech). The depleted RNA was polyadenylated with poly(A) polymerase (PAP) in the presence of a manganese catalyst, adding 50–90 adenosines per molecule. cDNA libraries were prepared with the Nanopore cDNA-PCR kit, pooled and sequenced on a PromethION device equipped with R10 flow cells (Oxford Nanopore Technologies).

**Long-read RNA sequencing data preprocessing.** For each sequenced strain, the following genome assemblies and gene annotations from NCBI RefSeq [ref] were used: GCF_000013425.1 (*Staphylococcus aureus RN450*), GCF_000069185.1 (*Mycobacterium abscessus ATCC 19977*), GCF_000195955.2 (ΔleuD ΔpanCD *Mycobacterium tuberculosis H37Rv 102J23*), GCF_000006765.1 - (*Pseudomonas aeruginosa PAO1*), and GCF_002899475.1 (*Escherichia coli DH5α*).

ONT reads from each replicate were polished with Pychopper (v2.7.10), polyA tails longer than 10 bases and sequencing adapters were trimmed using cutadapt and mapped against the genome assemblies using Minimap2 (v2.29)(Li, 2018) and Samtools (v1.22) (Danecek et al., 2021). Candidate operons were identified from read alignments spanning at least two genes on the same strand and then extended by combining overlapping candidates at most 50 base pairs apart. Operons were then collated from the triplicates for each strain and used as our operon annotations.

**Split.** We evaluate the operon identification in a zero-shot manner, therefore, we do not split the data into train, validation and test splits and use the entire dataset as a test set.

## A.3 PROTEIN-PROTEIN INTERACTION PREDICTION

We downloaded all the data from the STRING DB download site (https://string-db.org/cgi/download). Using the species metadata file we selected only bacterial organisms and downloaded the protein sequences for them together with protein-protein interaction scores for protein pairs. After running the download scripts we ended up with $10,533$ unique strains across 6,956 species. We used the *combined* interaction score which combines information from various sources to get a final score. STRING DB provides only protein sequences and no DNA, and the interaction scores are computed mainly at the protein-level, therefore, for this dataset we only provide protein sequences and omit DNA. To binarize the interaction scores, we set the threshold at $0.6$ ($\geq 0.6$=interaction, ¡0.6=no interaction). This threshold was chosen through conducting small-scale experiments and looking at the average performance of AUROC and AUPRC on the validation set across genomes, choosing the threshold which attains the best average performance across the two.

**Split.** We performed a random data split into training, validation, and test sets in a 70 / 10 / 20 % ratio. The larger proportion of the train set compared to the gene essentiality task is motivated by the larger size of the overall dataset, with the 10% validation set still allowing for meaningful evaluation.

## A.4 STRAIN CLUSTERING

To extract the metagenome assembled genomes (MAGs) for strain clustering, we use MGnify (Mitchell et al., 2023), which is a large-scale bacterial genomics database containing a diverse set of MAGs across numerous environments. The main reason for choosing MGnify over other potential resources is its large size combined with a uniform processing pipeline, providing comparable genomes. We wanted to evaluate whether various methods capture phylogenetic similarities across different taxonomic levels, therefore, we looked at strains which span different species, genera and families. These nested ranks provide three increasingly coarse resolutions to test whether an embedding preserves evolutionary signal. We use the taxonomic annotations provided by MGnify. The total number of unique genomes in the dataset is $6,071$.

To extract the genomes of interest, we extracted the most common species from MGnify from the corpus of 300k bacterial genomes and selected 25 species which are distributed across distinct genera and families. For a meaningful evaluation each genus (or family) must contain *at least two* species, otherwise the genus- or family-level clustering metrics become degenerate (every strain would be trivially assigned to a unique cluster at that rank). We download the chosen assemblies and use the accompanying annotations to translate gene DNA to protein sequences, while retaining the original DNA for the DNA-modality experiments.

**Split.** The strain clustering task is a fully unsupervised task, therefore, we do not split the data into train, validation and test splits and use the entire dataset as a test set.

### A.5 ANTIBIOTIC RESISTANCE PREDICTION

We leveraged the antibiotic sensitivity readings from the NCBI AST browser (`https://www.ncbi.nlm.nih.gov/pathogens/ast`), which contains hundreds of thousands of antibiotic-susceptibility test (AST) records for a diverse set of antibiotics. Using the genome assembly identifiers provided by the NCBI AST browser, we downloaded the DNA and protein sequences for each genome from the NCBI GenBank (`https://www.ncbi.nlm.nih.gov/genbank/`) and matched them with the antibiotic resistance readings. This left us with $26,052$ unique genomes with matched antibiotic resistance labels. We then processed the antibiotic resistance readings into *(i)* binary and *(ii)* regression labels decribed below.

**Binary.** For binary prediction, antibiotic sensitivity labels from NCBI AST browser of either *sensitive* (S) or *resistant* (R) were used for training and testing. If the antibiotic sensitivity test had no S/R label, they were not included. We remove antibiotics which 1) have less than $500$ available unique genomes in total, and 2) have less than $50$ unique genomes per class. This is motivated by the need to ensure that every classifier is trained on a sufficiently large and reasonably balanced data set; with fewer than $500$ genomes overall, or fewer than $50$ genomes in either class, the resulting model would suffer from poor statistical power and unreliable performance estimates. This resulted in 37 unique antibiotics. The Supp. Table 2 shows the number of available genomes per drug, including the number of susceptible and resistant genomes. The number of available readings varies strongly between antibiotics, which partly explains the high variance between the per drug performance.

**Regression.** For regression MIC prediction, NCBI AST browser quotes minimum inhibitory concentrations (MICs) as $< x$, $\leq x$, $= x$, $\geq x$ and $> x$. These strings were translated to an actual number ($y$) by $y = x$ if NCBI quoted MIC=$x$, $\leq x$ or $\geq x$; $y = 2 \times x$ if MIC quoted as MIC $> x$, and $y = 0.5 \times x$ if MIC was $< x$. We filter out antibiotics with less than $500$ available readings to ensure that the models are trained on a sufficiently large sample. This resulted in 56 antibiotics. To dampen the long tails, we normalized the MIC with a $log1p$ transformation. The final MIC distributions per antibiotic can be found in Supp. Fig 1 & 2.

**Split.** Due to low number of samples for many antibiotics and the variability between genomes, which may skew the results when using a single split, we trained and evaluated all models and antibiotics with $k$-fold split. Specifically, for each antibiotic we recommend: (1) splitting the available data into 5 equal splits using stratified split for the binary case and random split for regression. (2) In each split, further divide the larger train set into train and val, where validation makes up 20% of the train split. (3) Training the model on the train set and use the best performing model on the validation to evaluate the model on the test set.

Supplementary Table 2: Number of resistant, susceptible, and total labelled genomes for every antibiotic in the binary prediction setting.

| Drug | # Resistant | # Susceptible | # Total |
|---|---|---|---|
| amikacin | 7353 | 588 | 7941 |
| ampicillin | 10052 | 6211 | 16263 |
| amoxicillin–clavulanic acid | 12458 | 1683 | 14141 |
| azithromycin | 10497 | 2509 | 13006 |
| cefazolin | 1666 | 1897 | 3563 |
| cefepime | 3237 | 1188 | 4425 |
| cefotaxime | 566 | 446 | 1012 |
| cefoxitin | 12463 | 3902 | 16365 |
| ceftazidime | 2362 | 717 | 3079 |
| ceftriaxone | 13385 | 3469 | 16854 |
| ciprofloxacin | 17353 | 4280 | 21633 |
| clindamycin | 4327 | 337 | 4664 |
| ertapenem | 7708 | 373 | 8081 |
| erythromycin | 5605 | 1860 | 7465 |
| fosfomycin | 1186 | 396 | 1582 |
| gentamicin | 18242 | 2951 | 21193 |
| imipenem | 8485 | 409 | 8894 |
| kanamycin | 3768 | 410 | 4178 |
| levofloxacin | 3302 | 1106 | 4408 |
| meropenem | 11320 | 811 | 12131 |
| nalidixic acid | 1262 | 780 | 2042 |
| nitrofurantoin | 10535 | 5316 | 15851 |
| oxacillin | 6929 | 762 | 7691 |
| piperacillin–tazobactam | 3294 | 968 | 4262 |
| rifampin | 611 | 354 | 965 |
| streptomycin | 11965 | 2442 | 14407 |
| sulfamethoxazole | 3770 | 707 | 4477 |
| sulfisoxazole | 1569 | 323 | 1892 |
| tetracycline | 6936 | 3242 | 10178 |
| tigecycline | 662 | 357 | 1019 |
| trimethoprim | 2617 | 582 | 3199 |
| ampicillin–sulbactam | 1632 | 451 | 2083 |
| aztreonam | 2530 | 728 | 3258 |
| ceftaroline | 611 | 152 | 763 |
| chloramphenicol | 5345 | 1016 | 6361 |
| colistin | 572 | 290 | 862 |
| daptomycin | 492 | 131 | 623 |

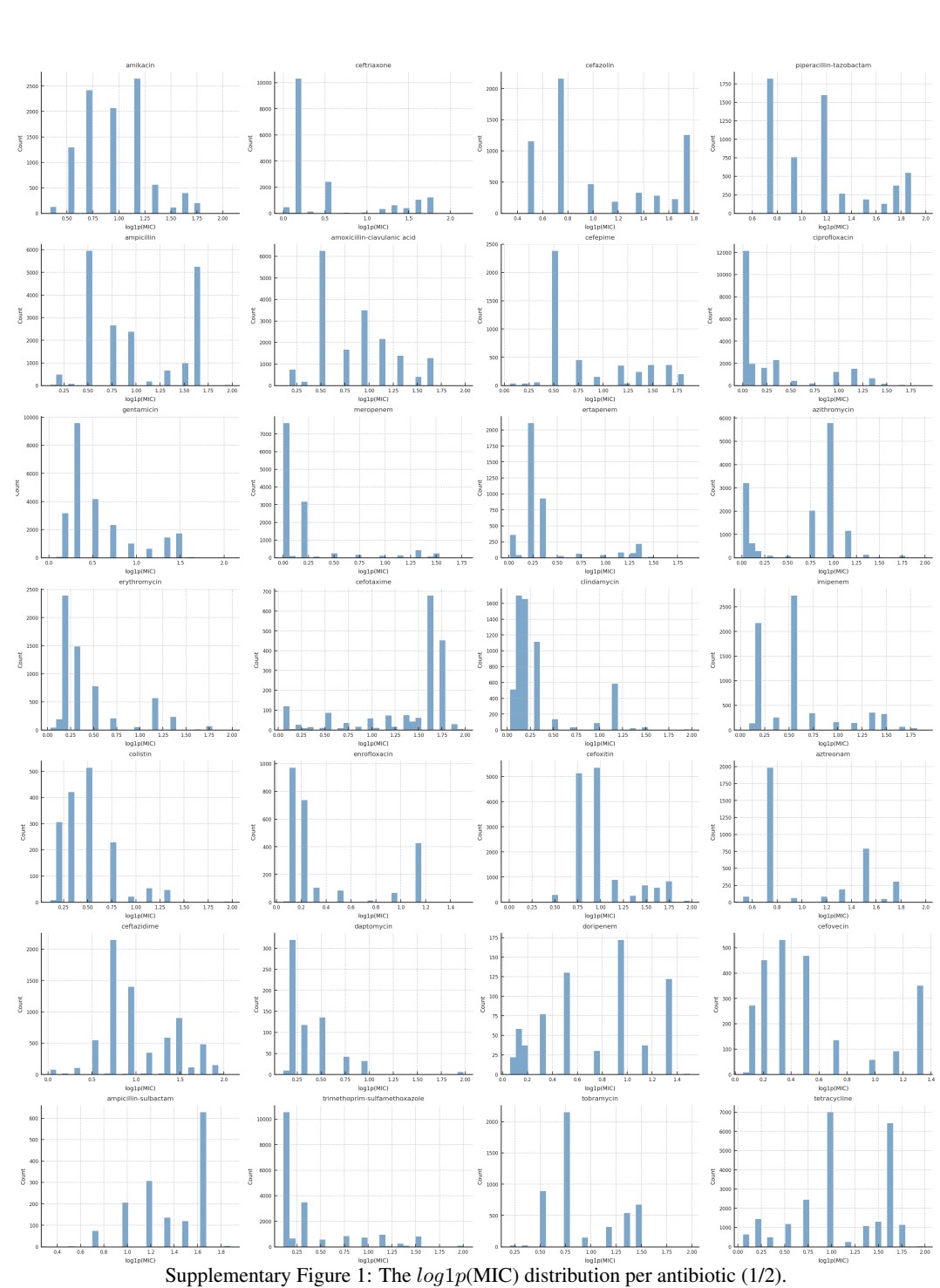

Supplementary Figure 1: The $log1p$(MIC) distribution per antibiotic (1/2).

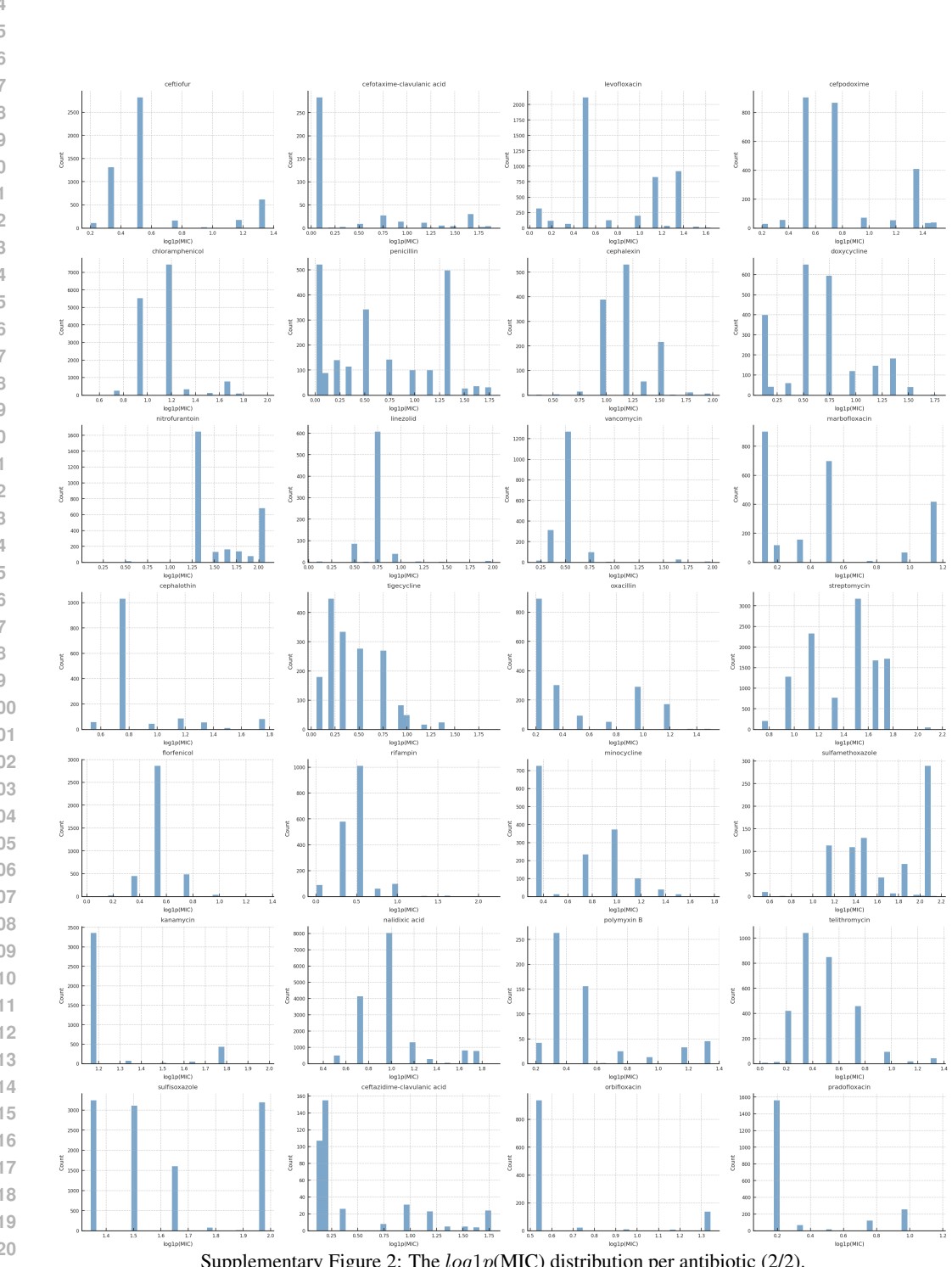

Supplementary Figure 2: The $log1p$(MIC) distribution per antibiotic (2/2).

## A.6 PHENOTYPIC TRAITS PREDICTION

We collected the phenotypic traits by collating two major sources (Madin et al., 2020; Weimann et al., 2016). To each phenotypic trait label we prepend its data source name so that the provenance of every label is explicit and any potential name collisions between the two catalogues are avoided. We keep duplicate traits that appear in both sources because they expand the number of labelled genomes without forcing us to merge measurements that were obtained with different experimental protocols. Using the taxonomy IDs and assembly accessions provided in the phenotypic traits datasets, we downloaded the associated genome DNA and protein sequences from the NCBI GenBank (`https://www.ncbi.nlm.nih.gov/genbank/`). We limit ourselves to categorical phenotypes and filter out the phenotypic traits with less than $500$ genomes to ensure that the models are trained and evaluated on a sufficiently large sample. Additionally, we removed the classes with less than $50$ samples. This resulted in $139$ unique phenotypes across $24,462$ genomes. We group the phenotypic traits into 5 distinct groups according to the type of biological information they capture (Supp. Fig. 3). We include the distributions of each phenotypic trait label in the Supp. Fig. 3-5 which shows large variation in the number of available labels per phenotypic trait which affects the final results.

**Split.** For each phenotype, we split the data into $60/20/20$ train, validation and test partitions respectively. As similar genomes often share the same phenotype, we split the data for each phenotype by genus—placing all genomes from a genus in one split—and evaluate on held-out genera, enforcing generalization to phylogenetically distant strains. To obtain stable estimates, as many traits are rare, we aggregate the per-phenotype results across 5 independent runs with different data splits.

Supplementary Table 3: Phenotype groups and their associated phenotypes. The name before the first "_" symbolizes the dataset source. *madin* phenotypes were extracted from Madin et al. (2020) and *gideon* from Weimann et al. (2016).

| Phenotype group | Phenotype |
|---|---|
| **Biochemical Activity** | gideon_Gelatin hydrolysis, gideon_Indole, gideon_Urea hydrolysis, gideon_Methyl red, gideon_VP (Voges Proskauer), gideon_-Gal (beta-galactosidase), gideon_Beta hemolysis, gideon_Hydrogen sulfide, gideon_Esculin hydrolysis |
| **Carbon Utilization** | madin_carbsubs_cellobiose, madin_carbsubs_glucose, madin_carbsubs_glycerol, madin_carbsubs_lactose, madin_carbsubs_maltose, madin_carbsubs_mannitol, madin_carbsubs_sucrose, madin_carbsubs_xylose, gideon_D-Arabitol, gideon_D-Mannose, gideon_Sucrose, gideon_D-Sorbitol |
| **Growth Conditions** | madin_categorical_range_tmp, madin_categorical_range_pH, madin_quantitative_optimum_tmp, madin_quantitative_optimum_pH, madin_quantitative_optimum_O2, madin_quantitative_growth_rate, gideon_Growth on ordinary blood agar, gideon_Growth on MacConkey agar |
| **Morphology** | madin_categorical_gram_stain, madin_categorical_sporulation, madin_categorical_cell_shape, madin_quantitative_average_cell_size, gideon_Motility, gideon_Spores, gideon_Shape: bacillus or coccobacillus, gideon_Branching filaments present |
| **Respiration Metabolism** | madin_categorical_metabolism, gideon_Nitrate reduction, gideon_Alanine aminopeptidase, gideon_O/F glucose oxidizer, gideon_Gas from glucose, gideon_Glucose fermenter, gideon_Aerobe, gideon_Oxidase, gideon_Facultative |

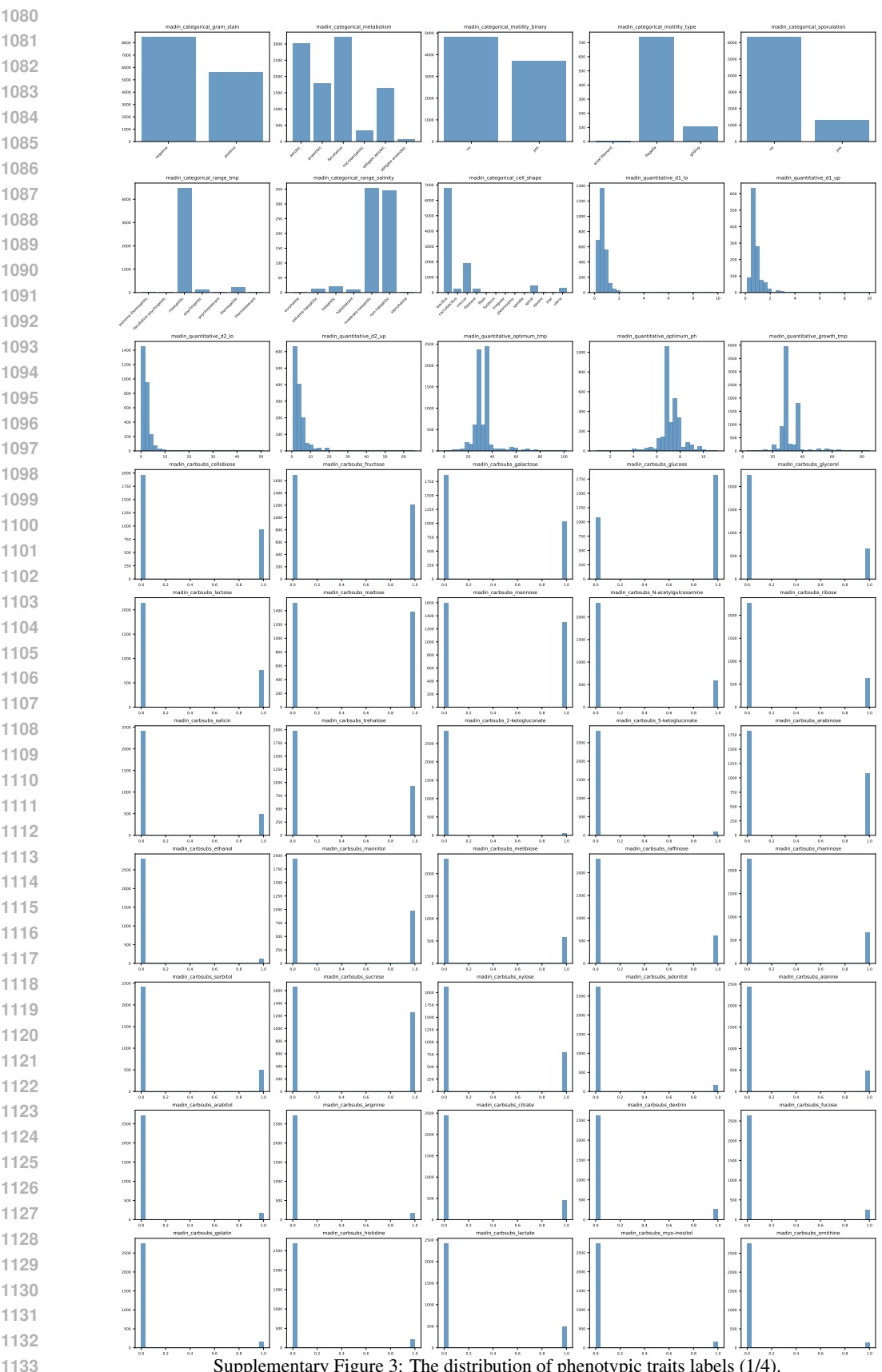

Supplementary Figure 3: The distribution of phenotypic traits labels (1/4).

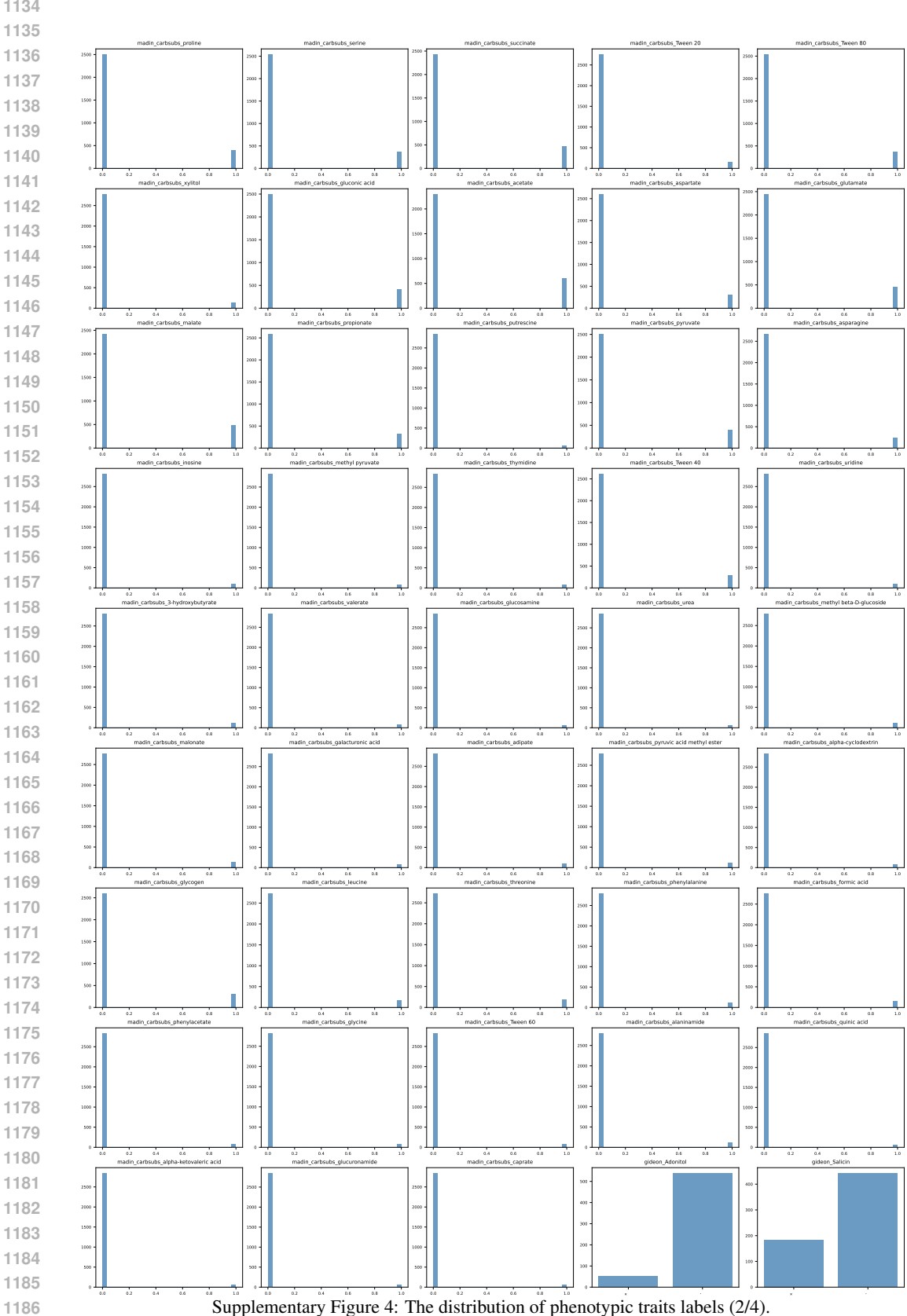

Supplementary Figure 4: The distribution of phenotypic traits labels (2/4).

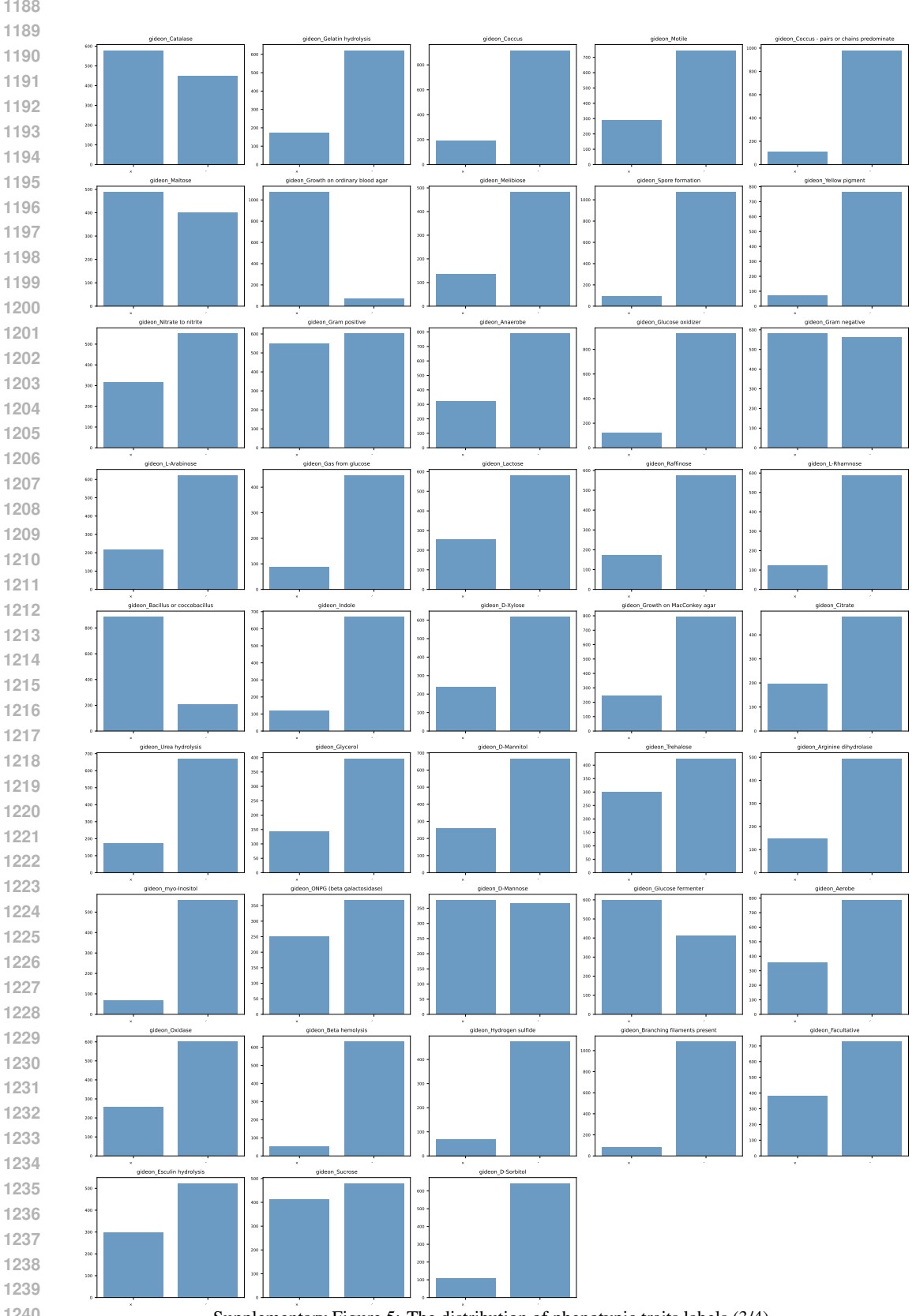

Supplementary Figure 5: The distribution of phenotypic traits labels (3/4).

1242
1243
1244
1245
1246
1247
1248
1249
1250
1251
1252
1253
1254
1255
1256
1257
1258
1259
1260
1261
1262
1263
1264
1265
1266
1267
1268
1269
1270
1271
1272
1273
1274
1275
1276
1277
1278
1279
1280
1281
1282
1283
1284
1285
1286
1287
1288
1289
1290
1291
1292
1293
1294
1295

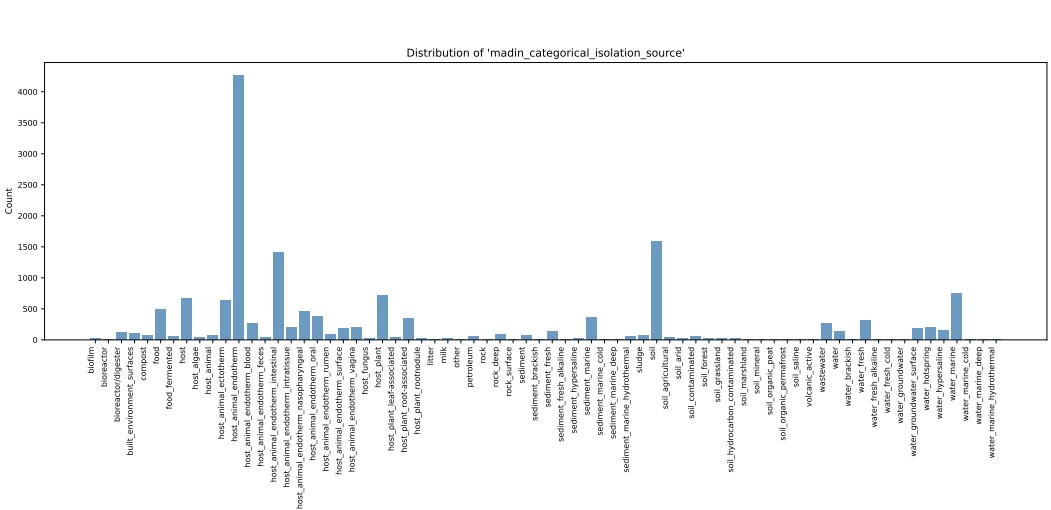

Supplementary Figure 6: The distribution of phenotypic traits labels (4/4).

## B  ADDITIONAL MODELING & EVALUATION DETAILS

We outline the experimental details used for modeling & evaluation for all model types; DNA Language Models (DNA LMs), protein Language Models (pLMs) and bacterial Language Models (bLMs). Implementation and further details can be found at `https://anonymous.4open.science/r/BacBench-B6EF`. All of the modeling code has been implemented in `PyTorch` (Paszke, 2019).

**DNA LMs.** For every model we load the public checkpoint (Supp. Table 4) and keep the hidden states of the *last* encoder layer. If a DNA string has length of $G$ tokens, the encoder produces $\mathbf{X} \in \mathbb{R}^{G \times D}$; we collapse it with a simple mean-pool to obtain a vector

$$\mathbf{z} = \frac{1}{G} \sum_{g=1}^{G} \mathbf{X}_{g,\cdot} \in \mathbb{R}^{D}.$$

**Gene- & system-level.** When the input gene (plus upstream promoter) is longer than the model context $C$, we slide a window of length $C$ with stride $s$ and embed each window. Averaging the $M$ window vectors $\{\mathbf{z}^{(1)}, \ldots, \mathbf{z}^{(M)}\}$ gives the final gene representation $\bar{\mathbf{z}}_{\text{gene}} = \frac{1}{M} \sum_m \mathbf{z}^{(m)}$.

**Genome-level.** We treat the whole genome in the same way: split into $C$-bp windows, embed each, and average; if several contigs are present we first average per-contig and then across contigs.

**pLMs.** A protein of $K$ amino acids yields $\mathbf{X} \in \mathbb{R}^{K \times D}$ and $\mathbf{z} = \frac{1}{K} \sum_k \mathbf{X}_{k,\cdot}$. For genome-level tasks we aggregate the $M$ protein vectors in that genome via $\bar{\mathbf{z}}_{\text{prot}} = \frac{1}{M} \sum_{i=1}^{M} \mathbf{z}_i$. Gene- and operon-level tasks use only the proteins involved.

**bLMs.** **gLM2** ingests *mixed-modality* genomic scaffolds in which protein-coding regions are translated to amino-acid tokens and intergenic regions remain as nucleotide tokens. We tokenize each scaffold into a single sequence up to the model context $C$ (4,096 tokens); longer scaffolds are processed with a sliding window of length $C$ and stride $s$, mirroring the DNA LM setup. For each window, we retain the last-layer hidden states and mean-pool to obtain a window vector $\mathbf{z}^{(m)} \in \mathbb{R}^{D}$; genome-level embeddings average across windows (and across contigs when present):

$$\bar{\mathbf{z}}_{\text{genome}} = \frac{1}{M} \sum_{m=1}^{M} \mathbf{z}^{(m)}.$$

For gene essentiality and operon tasks, inputs include contextual flanking sequence to supply regulatory/positional cues, but the gene embedding is computed by averaging only the hidden states of tokens that fall within the gene's coordinates; empirically this has shown to outperform averaging the entire slice.

**Bacformer** model takes as input *local* protein vectors $\mathbf{z}_1, \ldots, \mathbf{z}_M$ obtained exactly as in the pLM setting. They are then **ordered by their genomic coordinates** (chromosome followed by plasmids) so that the model can "see" genome organisation. Rotary positional embeddings (Su et al., 2024) are added to these vectors and the ordered sequence is fed to a *genome-level* Transformer encoder (Vaswani et al., 2017) with $L$ layers,

$$\mathbf{H}^{(0)} = [\mathbf{z}_1; \ldots; \mathbf{z}_M], \qquad \mathbf{H}^{(\ell+1)} = \text{Transformer}^{(\ell)}\big(\mathbf{H}^{(\ell)}\big), \; \ell = 0, \ldots, L-1.$$

The output $\mathbf{H}^{(L)} = [\tilde{\mathbf{z}}_1; \ldots; \tilde{\mathbf{z}}_M]$ contains **contextualised protein embeddings** $\tilde{\mathbf{z}}_i \in \mathbb{R}^{D}$ that encode both the protein sequence and its genomic neighbourhood. During pre-training Bacformer learns to predict which proteins co-evolved, so these embeddings capture functional coupling across the genome.

Empirically, averaging token embeddings performed slightly better than using the special `[CLS]` token, so mean pooling was used for every model throughout the paper.

### B.1  GENERAL TRAINING DETAILS.

For all tasks and settings which required finetuning except gene essentiality prediction, we kept the frozen backbone encoder model frozen, and only finetuned the neural network layer(s) stacked on

Supplementary Table 4: Summary of benchmarked models. "Max ctx." = maximum context length supported at inference; for DNA models measured in base pairs, for pLMs in amino acids, and for Bacformer in number of proteins present in the genome. "dim" = dimensionsionality of the output of the last hidden layer. *We use either `bacformer-masked-complete-genomes` or `bacformer-masked-MAG` depending on the genome type used as input. The DNA LMs, pLMs and bLMs are separated by a horizontal line.

| Model | Input | Variant / Checkpoint | Objective | Tokenisation | Params | dim | Training corpus | Max ctx. |
|---|---|---|---|---|---|---|---|---|
| Mistral-DNA (Mourad, 2025) | DNA | `Mistral-DNA-v1-138M-bacteria` | Autoregressive | Byte-pair | 138M | 768 | Bacteria | 512 |
| DNABERT-2 (Zhou et al., 2023) | DNA | `DNABERT-2-117M` | Masked | Byte-pair | 117M | 768 | Multi-kingdom | 512 |
| Nucleotide Transformer (Dalla-Torre et al., 2024) | DNA | `nucleotide-transformer-v2-250m-multi-species` | Masked | $k$-mer | 250M | 768 | Multi-kingdom | 2,048 |
| ProkBERT (Ligeti et al., 2024) | DNA | `neuralbioinfo/prokbert-mini-long` | Masked | $k$-mer | 27M | 384 | Bacteria | 4,096 |
| Evo (Nguyen et al., 2024) | DNA | `evo-1-8k-base (1.1_fix)` | Autoregressive | Single nucleotide | 6.5B | 4,096 | Multi-kingdom | 8,192 |
| ESM-2 (Lin et al., 2022) | Single protein seq. | `esm2_t12_35M_UR50D` | Masked | Single amino acid | 35M | 480 | Multi-kingdom | 1,024 |
| ESM-C (ESM Team, 2024) | Single protein seq. | `esmc_300m` | Masked | Single amino acid | 300M | 960 | Multi-kingdom | 1,024 |
| ProtBERT (Elnaggar et al., 2021) | Single protein seq. | `prot_bert` | Masked | Single amino acid | 420M | 1,024 | Multi-kingdom | 1,024 |
| gLM2 (Cornman et al., 2024) | Mixed modality (DNA & protein seq.) | `tattabio/gLM2_650M` | Masked | Single nucleotide/amino acid | 650M | 1,280 | Bacteria | 4,096 |
| Bacformer (Wiatrak et al., 2025) | Multiple protein seq. | `macwiatrak/bacformer-masked-complete-genomes*` | Masked | Single protein | 27M | 480 | Bacteria | 6,000 |

top of the model encoder. This was motivated by the computational cost required to embed all 67k genomes with all models. Further details on runtime and computational cost can be found below. We used Adam optimizer (Kingma, 2014) in all finetuning setups. All of the checkpoints used have been downloaded directly from HuggingFace and are specified in Supp. Table 4. For each task and model combination, we tuned the learning rate keeping other parameters unchanged. Further details on hyperparameters used for each task and setup can be found in task-specific sections below.

Supplementary Table 5: Model-specific context parameters used in this study. "Max ctx." is the maximum input length; "DNA-seq overlap" is the stride between consecutive windows when sliding across a genome; "Promoter length" is the upstream sequence length concatenated for promoter prediction.* For Bacformer the maximum input size of a protein is $1,024$ amino acids and maximum number of proteins in the genome is set to $6,000$.

| Model | Max ctx. | DNA-seq overlap | Promoter length |
|---|---|---|---|
| Mistral-DNA | 512 | 16 | 128 |
| DNABERT-2 | 512 | 16 | 128 |
| Nucleotide Transformer | 2,048 | 32 | 128 |
| ProkBERT | 4,096 | 64 | 128 |
| Evo | 8,192 | 32 | 128 |
| ProtBERT | 1,024 | N/A | N/A |
| ESM-2 | 1,024 | N/A | N/A |
| ESM-C | 1,024 | N/A | N/A |
| gLM2 | 4,096 | 64 | 128 |
| Bacformer | 1,024 / 6,000* | N/A | N/A |

## B.2 TASK DETAILS

We outline the experimental details for each task, outlining the hyperparameters used and evaluation setup. All of the experiments have been performed on a single NVIDIA A100 with 32 CPU cores.

**Gene essentiality prediction.** We stacked a single linear layer preceeded by a dropout of $0.2$ and layer normalization (Ba et al., 2016) on top of the gene embeddings and trained it with binary cross-entropy loss to predict gene essentiality (1=essential, 0=non-essential). We trained the model for maximum of 100 epochs with early stopping patience of 10, monitoring the macro AUROC across genomes on the validation set. For each model we tuned the learning rate specified in Supp. Table 6. The weight decay for the Adam optimizer has been set to $0.02$ for all models.

**Evo.** Evo natively does not provide straight-forward access to the output of the last hidden layer of the model. Therefore, we experimented with two ways of extracting the gene embeddings from Evo. Given a gene sequence $G$ of size $N$, 1) we used the script provided as part of the Evo implementation (https://github.com/evo-design/evo) to score the log-likelihoods of the nucleotides in a sequence, resulting in a vector $z \in \mathbb{R}^N$, and 2) modified the Evo model to return the output of the last hidden state, resulting in a matrix $X \in \mathbb{R}^{N \times D}$, where $D$ is model dimensionality which here equals $4,096$. We then similarly as with other models took the average of all sequence tokens resulting in a vector $x \in \mathbb{R}^{\mathbb{D}}$. We include this Evo implementation in the BacBench code repository (https://anonymous.4open.science/r/BacBench-B6EF). The option 1) yielded much better results on the validation set, therefore, we used it for final benchmarking.

Supplementary Table 6: Learning rates used for essential genes linear layer.

| Method | Learning rate |
|--------|---------------|
| Mistral-DNA | 0.005 |
| DNABERT-2 | 0.005 |
| Nucleotide Transformer | 0.005 |
| ProkBERT | 0.01 |
| Evo | 0.001 |
| ProtBERT | 0.005 |
| ESM-2 | 0.005 |
| ESM-C | 0.005 |
| gLM2 | 0.001 |
| Bacformer | 0.005 |

**Operon identification.** The operon–identification task is evaluated *zero-shot*; no fine-tuning is performed. We formulated operon prediction as a binary boundary classification problem, evaluating whether two contiguous genes form an operon. To address this, we developed a method that incorporates three features: (1) gene embeddings and (2) the strand of each gene.

First, we compute cosine similarity between adjacent genes using embeddings from the last hidden layer of pretrained models. We also record each gene's transcriptional strand from the genome assembly.

We then define a simple score that combines similarity and strand co-orientation:

$$s_i = c_i\, I_{\text{strand}}, \qquad c_i = \tfrac{1}{2}\big(1 + \cos(\hat{h}_i, \hat{h}_{i+1})\big),$$

where $I_{\text{strand}} = 1$ if both genes are on the same strand and 0 otherwise, and $\hat{h}_i, \hat{h}_{i+1}$ are the $\ell_2$-normalised embeddings of genes $i$ and $i+1$. The score $s_i \in [0,1]$ serves as the operon-membership score; a pair is classified as belonging to the same operon when $s_i$ exceeds a threshold. Mapping cosine similarity to $[0,1]$ stabilizes the scale across models, while the strand indicator provides a hard veto since genes on opposite strands do not belong to the same operon.

Performance is computed per strain.

**Evo.** Evo natively does not provide straight-forward access to the output of the last hidden layer of the model. Therefore, we experimented with two ways of extracting the gene embeddings from Evo. Given a gene sequence $G$ of size $N$, 1) we used the script provided as part of the Evo implementation (`https://github.com/evo-design/evo`) to score the log-likelihoods of the nucleotides in a sequence, resulting in a vector $z \in \mathbb{R}^N$, and 2) modified the Evo model to return the output of the last hidden state, resulting in a matrix $X \in \mathbb{R}^{N \times D}$, where $D$ is model dimensionality which here equals $4,096$. We then similarly as with other models took the average of all sequence tokens resulting in a vector $x \in \mathbb{R}^{\mathbb{D}}$. We include this Evo implementation in the BacBench code repository (`https://anonymous.4open.science/r/BacBench-B6EF`). The option 2) yielded significantly better results on operon identification, therefore, we used it for final benchmarking.

**Protein-protein interaction prediction.** To predict whether the two proteins interact, we fed the two protein embeddings into a linear model, which is trained to predict a binary label (1=interaction, 0=no interaction). The linear model is a single-layer neural network preceeded by a dropout of $0.2$ and layer normalization (Ba et al., 2016). The protein embeddings are fed into the linear classifier, after which the pairs of interacting proteins are averaged and passed through a final binary classification layer preceeded by a dropout of $0.2$. The model is trained to minimize the binary cross-entropy loss. We experimented with different learning rates for all the models and set the final learning rate to $0.001$, which has shown to perform the best for all the models. We trained the model for the maximum epochs of 10 and no early stopping patience. We set the maximum gradient norm to $2.0$ and monitor the validation loss across genomes. The weight decay for the Adam optimizer is set to $0.01$.

**Strain clustering.** To compute the strain-clustering metrics we run Leiden clustering (Traag et al., 2019) over a grid of parameters. We vary the *resolution* in `[0.1, 0.25, 0.5, 1.0]` and the

*number of neighbours* in `[5, 10, 15]`, evaluating every pairwise combination. Lower resolutions are omitted to avoid collapsing many genomes into a single (or just a few) giant clusters. After computing the clustering metrics for every parameter pair, we keep for each method the combination that maximises the mean performance across species-, genus- and family-level labels. The Leiden clustering is performed using the `scanpy` package (Wolf et al., 2018).

**Antibiotic resistance prediction.** To predict the antibiotic resistance, we train a linear model for each drug and method combination. The linear model is a single-layer neural network preceeded by a dropout of $0.2$ and layer normalization (Ba et al., 2016). We train the models separately for the *(i)* binary and *(ii)* regression MIC prediction case. We optimize the former for the binary cross entropy loss and the latter for the mean squared error loss. We train all models for the maximum of $100$ epochs with early stopping patience of $5$, monitoring the validation AUPRC in the binary setup and validation $R^2$ in the regresssion setup. We experimented with various learning rates for each model, setting the final learning rate to $0.005$ which attained the best results across folds and seeds for all models. We set the weight decay in the Adam optimizer to $0.01$.

We have also experimented with training a multi-task linear model, which simultaneously predicts antibiotic resistance of a genome to multiple drugs, however, it performed worse then a separate linear model for each drug.

**Phenotypic traits prediction.** To predict a phenotypic trait from a genome-level embedding, we train an linear model for each phenotype and method combination. The linear model is a single-layer neural network preceeded by a dropout of $0.2$ and layer normalization (Ba et al., 2016). As all labels are categorical, we optimize it to minimize the cross-entropy loss. We set the maximum number of epochs to $2,000$ and early stopping patience to $50$, monitoring the validation loss. The learning rate for all models was set to $0.01$. We use the cross-entropy loss. To account for the class imbalance, we weigh each class according to:

$$w_c = \frac{N}{K\,n_c},\tag{1}$$

where $n_c$ is the number of training samples in class $c$, $K$ is the total number of classes, and $N$ is the total number of training samples.

**Runtime analysis.** A typical bacterial genome contains on the order of $3{,}000-5{,}000$ genes and $\sim 4-6$ Mbp of DNA, with average protein lengths of $\sim 300$ amino acids; embedding an entire genome therefore entails thousands of forward passes for protein LMs and millions of tokens for DNA LMs, making raw throughput a practical bottleneck for population-scale studies. We measured the wall-clock time required to embed the *11* genomes used in the operon identification task on a single NVIDIA A100 and extrapolated to BacBench's full collection of $67{,}000$ genomes (Table 7). The fastest models are **ESM-2** and **Bacformer** (64–67 s for 11 genomes; $\approx$1.2–1.25k GPU-hours for 67k genomes). DNA LMs add a modest cost (e.g., 93 s for Mistral-DNA; 168 s for DNABERT-2) yet remain tractable on a single GPU. In stark contrast, the 6.5B-parameter **Evo** is $\sim 10^4\times$ slower ($5.76\times10^5$ s for 11 genomes, i.e., $\sim$14 h per genome), yielding an impractical $\sim 1.07\times10^7$ GPU-hours for 67k genomes on one GPU. These measurements underline that—even when accuracy is the primary goal—runtime quickly becomes the limiting factor for population-scale analyses.

Beyond embedding costs, fine-tuning on genome-level tasks requires backpropagating through *all* proteins per genome (median $\sim$2,506 proteins across our datasets) or through multi-megabase DNA contexts, which dramatically amplifies memory and compute, requiring often ¿200 NVIDIA A100 GPUs for a single genome. Practically, this makes genome-scale fine-tuning out of reach for most academic labs for DNA LMs and pLMs, while remaining feasible for Bacformer-style bLMs; thus, linear-probe evaluations are a necessary, controlled proxy for model selection at scale.

### B.3 DATASETS & MODELS LICENSES

To ensure that our datasets and benchmarks are reusable in the academic setting, we checked the license for each resource used in the manuscript. The Supp. Table 8 details a license for all models and datasets used in the manuscript.

Supplementary Table 7: Embedding runtime (s=seconds) on the operon identification task (11 genomes) on one NVIDIA A100 GPU and the extrapolated wall-clock time (h=hours) to process the entire collection of 67k bacterial genomes on the same hardware.

| Model | Runtime (s) | Estimated time for 67 k genomes (h) |
|---|---|---|
| Mistral-DNA | 93 | 1,731 |
| DNABERT-2 | 168 | 3,127 |
| Nucleotide Transformer | 100 | 1,861 |
| ProkBERT | 106 | 1,973 |
| Evo | 575,629 | 10,712,540 |
| ProtBERT | 95 | 1,768 |
| ESM-2 | 64 | 1,191 |
| ESM-C | 137 | 2,550 |
| gLM2 | 182 | 3,387 |
| Bacformer | 67 | 1,247 |

Supplementary Table 8: External resources used in this study and their licences.

| Resource | Type | Licence |
|---|---|---|
| ESM-2 | Model | MIT |
| ESM-C | Model | Cambrian Open License Agreement |
| ProtBERT | Model | Academic Free License v. 3.0 |
| gLM2 | Model | Apache 2.0 |
| Bacformer | Model | Apache 2.0 |
| DNABERT-2 | Model | Apache 2.0 |
| ProkBERT | Model | MIT |
| Evo | Model | Apache 2.0 |
| Nucleotide Transformer | Model | CC BY-NC-SA 4.0 |
| Mistral-DNA | Model | Apache 2.0 |
| MGnify | Data | CC0 1.0 Universal |
| NCBI AST Browser | Data | Public domain (U.S. Gov data) |
| NCBI GenBank/RefSeq | Data | Public domain (U.S. Gov data) |
| Database of Essential Genes | Data | CC BY 4.0 |
| Operon DB | Data | CC BY 4.0 |
| STRING DB | Data | CC BY 4.0 |

# C  ADDITIONAL RESULTS

We show and discuss further results across all tasks. To allow for comparability in future work, we include tables with all numerical results as well as per antibiotic and phenotypic traits scores.

## C.1  GENE ESSENTIALITY PREDICTION

The results on AUPRC show the bLMs and pLMs outperforming DNA LMs (Supp. Fig 7). gLM2 performs the best, showing the benefits of taking as input both DNA as well as protein sequence information. Moreover, increasing the pLM size appears to further boost performance, as demonstrated by ESM-C (300M) and ProtBERT (420M) outperforming ESM-2 (35M). Bacformer outperforms its protein representation backbone ESM-2, showing the benefits of incorporating whole-genome context. Evo outperforms other DNA LMs, except ProkBERT, demonstrating the performance gain by conducting pretraining on a relevant corpus. Finally the results on AUPRC show that there is large room for improvement. We believe that increasing the number and diversity of annotated genomes would significantly boost model performance.

The Supp. Table 9 shows the exact results across AUROC and AUPRC measured across disinct genomes.

**Finetuning performance.**  To further analyse model performance, we have conducted finetuning on the gene essentiality task. The results show that finetuning boosts performance (Table 10); however, the model ranking remains largely unchanged, with the pLMs and bLMs outperforming DNA LMs. We have excluded Evo from finetuning due to the computational complexity required to finetune

Evo (Table 7). Gene essentiality is the only task for which we finetuned all models; genome-level tasks such as antibiotic-resistance prediction require end-to-end, whole-genome context and are prohibitively expensive for all models except Bacformer—underscoring the need for models that can be finetuned efficiently for whole-genome tasks.

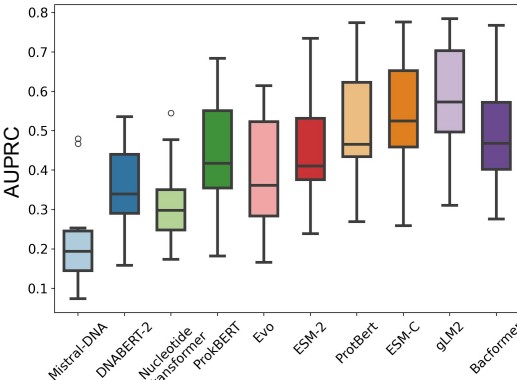

Supplementary Figure 7: AUPRC across test genomes on the gene–essentiality prediction task. The box spans the inter-quartile range with a line marking the median value.

Supplementary Table 9: Overall performance on gene–essentiality prediction. Values are *mean ± standard deviation* over 3 random seeds; the best score for each metric is highlighted in bold.

| Method | AUROC | AUPRC |
|---|---|---|
| Mistral-DNA | $57.52 \pm 2.86$ | $22.69 \pm 14.21$ |
| DNABERT-2 | $68.21 \pm 7.17$ | $35.76 \pm 12.46$ |
| Nucleotide Transformer | $67.03 \pm 5.56$ | $31.78 \pm 11.88$ |
| ProkBERT | $74.79 \pm 6.22$ | $44.79 \pm 15.25$ |
| Evo | $73.71 \pm 7.10$ | $39.08 \pm 15.21$ |
| ESM-2 | $77.99 \pm 5.82$ | $46.23 \pm 15.87$ |
| ESM-C | $82.25 \pm 6.04$ | $55.08 \pm 15.83$ |
| ProtBERT | $80.72 \pm 5.85$ | $52.00 \pm 15.90$ |
| gLM2 | $\mathbf{83.77 \pm 6.17}$ | $\mathbf{58.61 \pm 14.96}$ |
| Bacformer | $80.72 \pm 5.87$ | $50.33 \pm 15.43$ |

Supplementary Table 10: Overall finetuning performance on gene–essentiality prediction. Values are *mean ± standard deviation* over 3 random seeds; the best score for each metric is highlighted in bold.

| Method | AUROC | AUPRC |
|---|---|---|
| **Mistral-DNA** | $62.18 \pm 7.87$ | $28.81 \pm 13.51$ |
| **DNABERT-2** | $74.88 \pm 16.75$ | $56.12 \pm 29.71$ |
| **Nucleotide Transformer** | $73.06 \pm 14.99$ | $48.56 \pm 28.65$ |
| **ProkBERT** | $69.88 \pm 8.60$ | $44.98 \pm 14.36$ |
| **ESM-2** | $85.36 \pm 9.12$ | $64.42 \pm 17.79$ |
| **ESM-C** | $89.96 \pm 7.53$ | $71.85 \pm 14.49$ |
| **ProtBERT** | $\mathbf{91.31 \pm 73.02}$ | $73.02 \pm 16.56$ |
| **gLM2** | $90.12 \pm 7.87$ | $\mathbf{74.03 \pm 21.70}$ |
| **Bacformer** | $90.31 \pm 7.40$ | $72.13 \pm 15.16$ |

## C.2 OPERON IDENTIFICATION

The AUPRC follows the same trend as the AUROC described in the main manuscript: Bacformer attains the best scores, ProkBERT ranks second, and the remaining DNA LMs and pLMs form a tight cluster with broadly comparable performance—reinforcing that genome-level context and bacteria-specific pretraining matter more than modality alone; absolute AUPRC values are lower (as expected under class imbalance) but track the same model ordering.

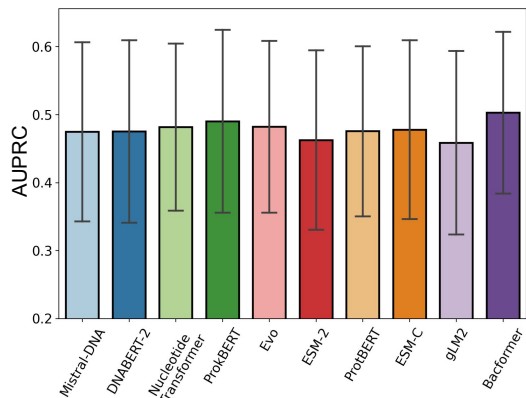

Supplementary Figure 8: AUPRC across test genomes on gene essentiality prediction task. The box spans the inter-quartile range with a line marking the median value.

Supplementary Table 11: Overall performance on operon identification. Values are *mean ± standard deviation*; the best score for each metric is highlighted in bold.

| Method | AUROC | AUPRC |
|---|---|---|
| Mistral-DNA | 74.16 ± 9.70 | 47.48 ± 29.47 |
| DNABERT-2 | 74.13 ± 10.14 | 47.53 ± 30.07 |
| Nucleotide Transformer | 74.50 ± 8.57 | 48.19 ± 27.47 |
| ProkBERT | 75.77 ± 9.69 | 49.03 ± 30.07 |
| Evo | 74.77 ± 8.33 | 48.21 ± 28.27 |
| ESM-2 | 73.02 ± 10.30 | 46.27 ± 29.55 |
| ESM-C | 75.25 ± 9.26 | 47.79 ± 29.39 |
| ProtBERT | 75.11 ± 8.65 | 47.58 ± 27.98 |
| gLM2 | 72.85 ± 10.77 | 45.86 ± 30.18 |
| **Bacformer** | **77.59 ± 6.64** | **50.31 ± 26.61** |

## C.3 PROTEIN-PROTEIN INTERACTION PREDICTION

The protein-protein interaction (PPI) prediction results on AUPRC (Supp. Fig. 9) show that contextual pLM, Bacformer, consistently outperforms other methods. We credit it to its usage of the genomic context. Moreover, the performance does not increase by scaling the model size, as shown by ESM-C and ProtBERT underperforming ESM-2. The overall results (Supp. Table 12) show relatively low performance considering the trainins set size (¿7k genomes) even in the finetuned setting. We believe this is due to the 1) complexity of the task, 2) noisy source data, as STRING DB where the interactions have been extracted from collates information from a variety of sources and is not limited to experimentally validated interactions, highlighting the importance of building high quality PPI datasets.

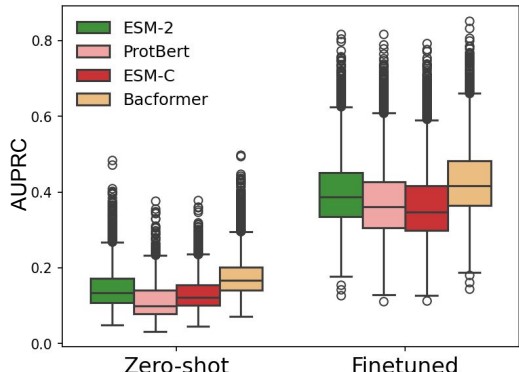

Supplementary Figure 9: AUPRC across test genomes on the protein–protein-interaction task in both zero-shot and fine-tuned settings. The box spans the inter-quartile range with a line marking the median.

Supplementary Table 12: Protein–protein-interaction prediction on the held-out genomes. Values are *mean ± standard deviation* over five seeds; the best score for each metric is shown in bold.

| Method | Zero-shot | | Finetuned | |
|---|---|---|---|---|
| | AUROC | AUPRC | AUROC | AUPRC |
| ESM-2 | $56.46 \pm 2.10$ | $14.94 \pm 6.04$ | $76.62 \pm 2.35$ | $40.68 \pm 10.71$ |
| ProtBERT | $47.20 \pm 3.88$ | $11.45 \pm 4.89$ | $74.05 \pm 2.53$ | $38.15 \pm 11.16$ |
| ESM-C | $55.17 \pm 2.66$ | $13.33 \pm 4.61$ | $74.21 \pm 2.37$ | $37.30 \pm 10.98$ |
| Bacformer | $\mathbf{63.09 \pm 2.73}$ | $\mathbf{18.20 \pm 6.28}$ | $\mathbf{79.09 \pm 2.25}$ | $\mathbf{43.47 \pm 10.61}$ |

## C.4 STRAIN CLUSTERING

In addition to the strain clustering metrics included in the main manuscript, we plotted 2-dimensional UMAP results colored by species (Supp. Fig. 10), genus (Supp. Fig. 11) and family (Supp. Fig. 12) for a subset of models for further investigation. The UMAPs show that the strain representations differ between the models. All models cluster strains into separate species clusters, however, not all of them retain the phylogenetic similarities between species in the same genus or family. We also notice that the DNA LMs tend to output less overlapping species clusters, which boosts its performance at species level, but leads to lower results at higher taxonomic levels (genus and family).

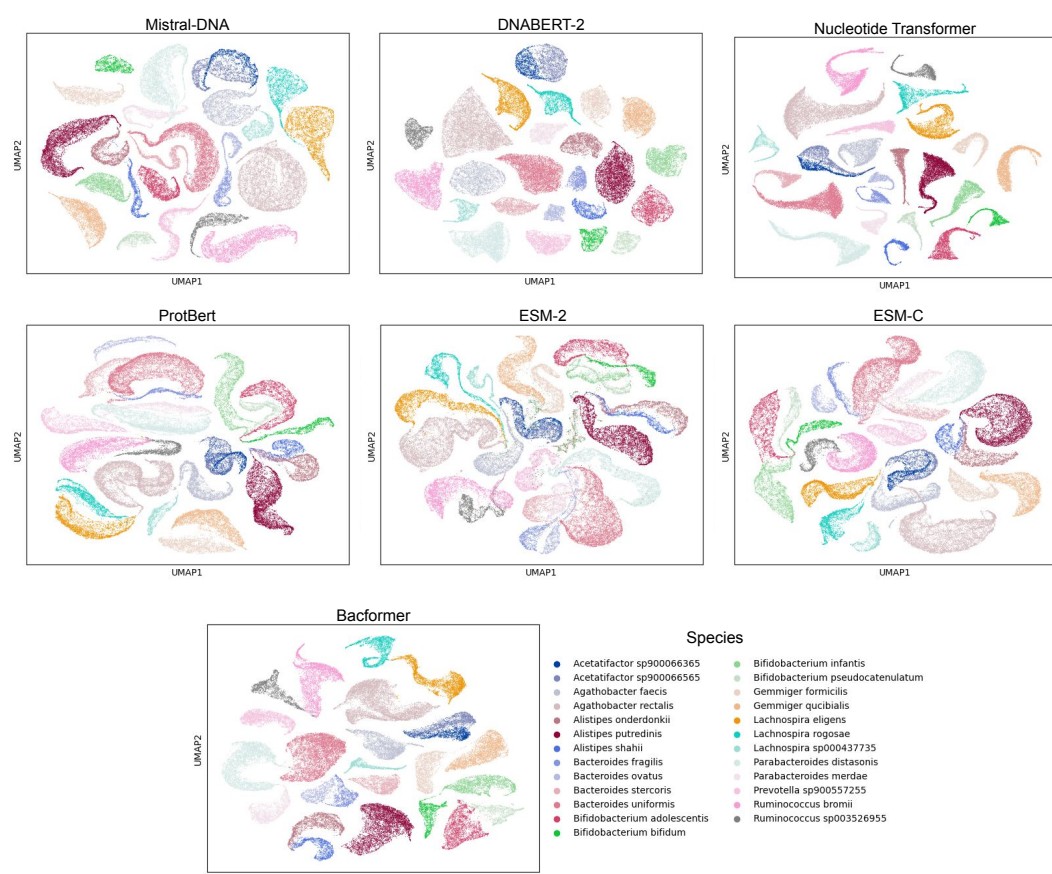

Supplementary Figure 10: UMAP plots of strains (i.e. genome-level embeddings) across diverse methods colored by species.

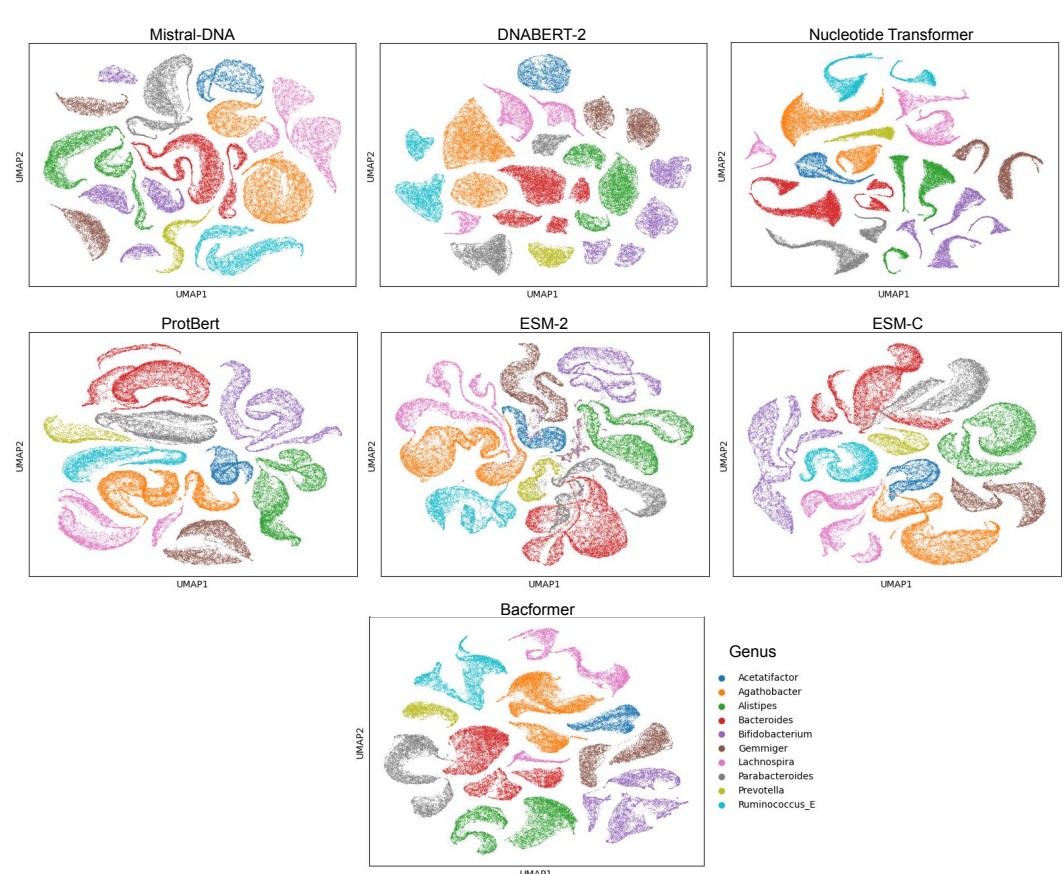

Supplementary Figure 11: UMAP plots of strains (i.e. genome-level embeddings) across diverse colored by genus.

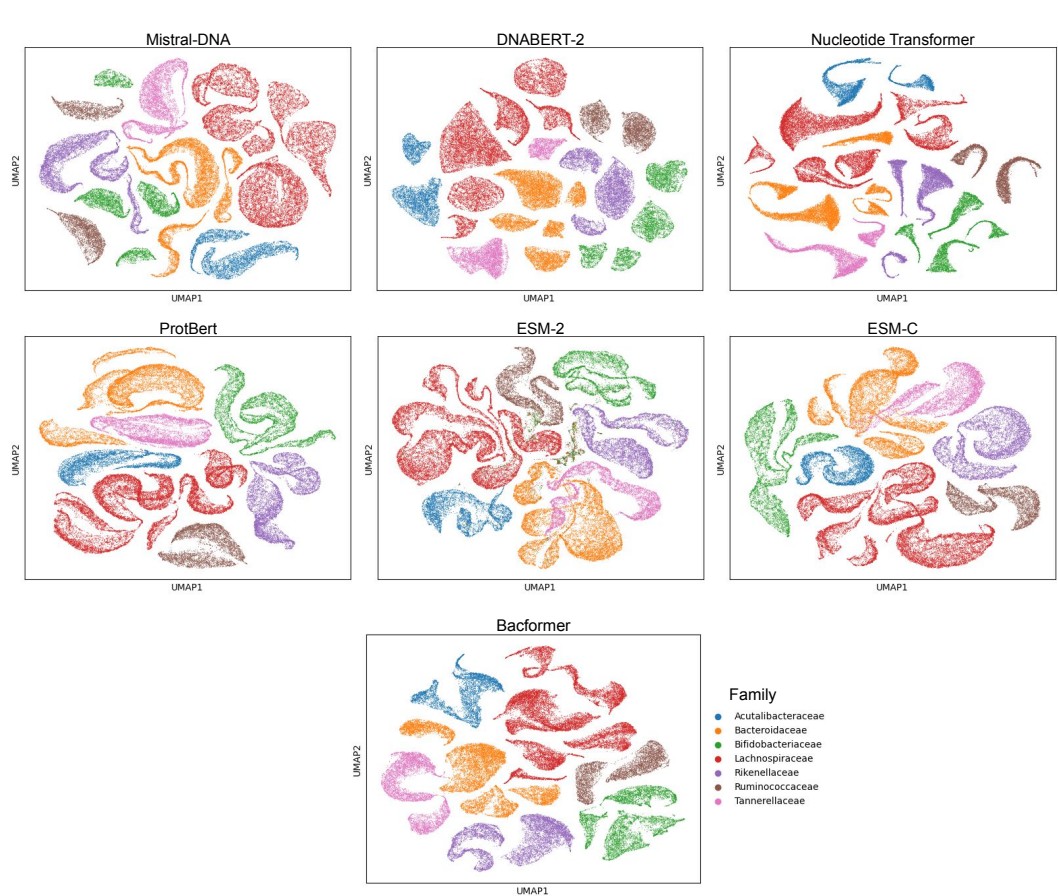

Supplementary Figure 12: UMAP plots of strains (i.e. genome-level embeddings) across diverse methods colored by family.

## C.5 ANTIBIOTIC RESISTANCE PREDICTION

On the following pages we present the results per antibiotic across metrics. This includes binary prediction results (susceptible/resistant, Supp. Table 13) and MIC regression prediction results (Supp. Table 14). Each antibiotic was run across 5-folds with 3 random seeds to avoid variance stemming from random initialization and data-split bias. In the binary prediction setting the bLM-Bacformer outperforms other methods, achieving the best AUROC on 26 drugs, AUPRC on 27 drugs and F1 on 24 drugs out of 37 in total. Thus, showcasing the benefits of considering the interactions between proteins present in the genome. In the regression setting, Bacformer attains the best result on 44 drugs on $R^2$ and 45 on *Pearson* correlation coefficient out of total of 56 antibiotics. Finally, we see large variance across as well as within drugs. The former can be partly explained by the variable number of labels available per antibiotic, while the latter shows that, even within a single antibiotic, model performance can fluctuate markedly across folds and random seeds—highlighting sensitivity to sample composition and pointing to the need for larger, more balanced datasets or stronger regularisation to obtain stabler estimates.

Supplementary Table 13: Per-antibiotic binary antibiotic resistance prediction. Values are mean ± standard deviation across 3 seeds. Bold indicates the highest mean for each metric.

| Antibiotic | Mistral-DNA | | DNABERT-2 | | Nucleotide Transformer | | ProkBERT | | ESM-2 | | ESM-C | | ProtBERT | | gLM2 | | Barformer | |
|---|---|---|---|---|---|---|---|---|---|---|---|---|---|---|---|---|---|---|
| | AUROC | AUPRC | AUROC | AUPRC | AUROC | AUPRC | AUROC | AUPRC | AUROC | AUPRC | AUROC | AUPRC | AUROC | AUPRC | AUROC | AUPRC | AUROC | AUPRC |
| amikacin | 90.24 ± 1.09 | 33.09 ± 1.79 | 88.46 ± 6.35 | 33.23 ± 9.22 | 90.77 ± 1.07 | 34.84 ± 3.83 | 90.47 ± 0.46 | 30.44 ± 0.30 | 91.62 ± 0.47 | 41.33 ± 0.75 | 91.94 ± 0.94 | 48.04 ± 3.46 | 91.70 ± 2.55 | 49.59 ± 6.31 | 88.94 ± 1.87 | 40.32 ± 2.48 | **94.43 ± 1.12** | **59.00 ± 4.79** |
| amoxicillin-clavulanic acid | 69.69 ± 1.80 | 28.06 ± 1.91 | 76.07 ± 1.53 | 37.21 ± 3.28 | 76.99 ± 2.30 | 41.06 ± 6.41 | 78.61 ± 0.50 | 42.47 ± 1.64 | 82.51 ± 1.63 | 48.70 ± 3.51 | 85.17 ± 1.03 | 54.77 ± 4.38 | 80.40 ± 1.07 | 48.71 ± 4.38 | 78.33 ± 1.48 | 42.78 ± 3.39 | **85.86 ± 0.73** | **59.63 ± 1.54** |
| ampicillin | 74.49 ± 1.42 | 68.27 ± 1.68 | 74.99 ± 1.32 | 69.81 ± 2.23 | 78.74 ± 0.94 | 72.00 ± 1.70 | 78.11 ± 0.47 | 71.77 ± 1.40 | 79.42 ± 0.24 | 73.05 ± 0.70 | 82.67 ± 0.59 | 77.14 ± 0.88 | 80.23 ± 0.47 | 74.55 ± 0.35 | 78.97 ± 0.67 | 71.77 ± 1.40 | **82.72 ± 0.84** | **77.85 ± 0.98** |
| ampicillin-subactam | 75.17 ± 4.69 | 77.76 ± 5.53 | 78.75 ± 4.65 | 79.29 ± 5.24 | 82.87 ± 3.82 | 82.53 ± 6.19 | 83.21 ± 4.42 | 81.04 ± 5.30 | 78.52 ± 4.68 | 81.63 ± 4.88 | 78.37 ± 3.48 | 81.83 ± 4.31 | 77.69 ± 6.97 | 81.15 ± 6.74 | 78.65 ± 3.30 | 78.09 ± 4.95 | **86.72 ± 5.01** | **88.06 ± 5.39** |
| azithromycin | 73.61 ± 7.84 | 3.57 ± 2.52 | 80.34 ± 3.02 | 7.31 ± 5.64 | 83.44 ± 0.90 | 9.15 ± 5.02 | 83.89 ± 1.14 | 4.62 ± 2.77 | 82.37 ± 6.40 | 8.08 ± 4.63 | 83.74 ± 5.03 | 7.91 ± 2.87 | 83.64 ± 2.58 | 7.30 ± 2.63 | 68.52 ± 12.37 | 6.76 ± 8.30 | **83.00 ± 3.73** | **9.86 ± 3.80** |
| aztreonam | 90.11 ± 0.68 | 90.05 ± 0.97 | 88.46 ± 1.67 | 89.37 ± 1.53 | 93.44 ± 0.90 | 89.63 ± 2.47 | 93.89 ± 1.14 | 93.53 ± 1.09 | 95.30 ± 0.39 | 94.25 ± 0.74 | **95.76 ± 0.37** | **94.74 ± 0.56** | 93.73 ± 0.89 | 93.40 ± 0.99 | 92.54 ± 0.65 | 92.28 ± 0.65 | 93.91 ± 1.21 | 93.09 ± 0.90 |
| cefazolin | 78.99 ± 1.58 | 83.03 ± 0.82 | 81.08 ± 1.32 | 84.62 ± 0.78 | 85.30 ± 2.61 | 51.49 ± 6.06 | 85.18 ± 2.72 | 90.00 ± 2.15 | 88.81 ± 3.86 | 90.00 ± 2.15 | **92.23 ± 1.25** | **94.13 ± 1.11** | 86.01 ± 0.51 | 90.29 ± 0.50 | 86.99 ± 0.26 | 90.97 ± 0.34 | 90.68 ± 1.24 | 93.61 ± 0.86 |
| cefepime | 71.17 ± 3.51 | 42.30 ± 3.73 | 71.83 ± 4.55 | 44.05 ± 4.25 | 75.30 ± 2.94 | 51.49 ± 6.06 | 76.21 ± 2.79 | 52.50 ± 2.25 | 79.50 ± 5.92 | 57.81 ± 12.81 | 82.08 ± 2.32 | 65.81 ± 4.49 | 78.94 ± 3.77 | 63.77 ± 5.14 | 76.11 ± 1.09 | 52.36 ± 2.19 | **84.75 ± 2.00** | **68.73 ± 5.33** |
| cefotaxime | 56.87 ± 1.73 | 79.94 ± 0.25 | 66.74 ± 3.36 | 89.85 ± 2.20 | 84.73 ± 3.33 | 95.39 ± 1.35 | 84.76 ± 4.79 | 95.39 ± 1.35 | 91.83 ± 2.39 | 97.54 ± 1.26 | 92.32 ± 2.25 | 97.78 ± 1.05 | 90.98 ± 2.98 | 97.36 ± 1.35 | 84.63 ± 2.64 | 95.96 ± 0.69 | 91.87 ± 2.58 | **97.82 ± 0.89** |
| cefoxitin | 76.41 ± 1.43 | 53.21 ± 0.34 | 80.23 ± 3.29 | 56.91 ± 4.44 | 82.94 ± 2.25 | 64.42 ± 2.09 | 82.10 ± 1.94 | 60.50 ± 1.73 | 85.36 ± 1.20 | 65.00 ± 1.26 | 87.56 ± 0.70 | 68.87 ± 1.68 | 84.79 ± 2.05 | 63.05 ± 5.57 | 81.91 ± 1.59 | 61.63 ± 1.07 | **89.03 ± 0.97** | **72.49 ± 2.16** |
| ceftazidime | 72.74 ± 2.12 | 60.18 ± 1.92 | 72.75 ± 1.58 | 60.63 ± 1.92 | 79.65 ± 0.63 | 69.71 ± 1.64 | 80.47 ± 0.61 | 71.54 ± 1.28 | 78.73 ± 5.68 | 68.29 ± 8.95 | 83.93 ± 0.26 | 75.92 ± 0.87 | 79.18 ± 1.04 | 73.29 ± 1.29 | 78.07 ± 2.83 | 66.86 ± 4.36 | **86.26 ± 0.30** | **80.74 ± 0.92** |
| ceftiofur | 74.81 ± 2.82 | 27.04 ± 3.63 | 81.13 ± 1.89 | 35.06 ± 5.20 | 86.11 ± 1.15 | 64.42 ± 2.09 | 82.75 ± 0.22 | 42.97 ± 3.83 | 81.77 ± 5.89 | 46.11 ± 10.96 | 86.39 ± 2.72 | 52.22 ± 9.21 | 81.33 ± 3.42 | 41.22 ± 9.21 | 77.05 ± 0.85 | 30.72 ± 1.76 | **88.73 ± 1.10** | 58.91 ± 3.37 |
| ceftriaxone | 74.60 ± 4.16 | 50.22 ± 3.72 | 74.00 ± 1.38 | 50.67 ± 2.37 | 85.10 ± 0.46 | 67.21 ± 0.77 | 80.76 ± 0.42 | 58.02 ± 0.79 | 87.59 ± 0.40 | 69.16 ± 1.52 | **90.51 ± 0.26** | **75.08 ± 0.75** | 81.72 ± 1.18 | 58.92 ± 1.22 | 77.03 ± 2.20 | 66.11 ± 0.97 | 88.98 ± 0.25 | 72.57 ± 0.65 |
| chloramphenicol | 72.18 ± 5.04 | 28.68 ± 4.05 | 75.99 ± 5.16 | 17.39 ± 1.97 | 83.29 ± 2.96 | 30.40 ± 9.12 | 83.25 ± 2.36 | 23.86 ± 5.20 | 84.62 ± 1.46 | 30.79 ± 3.68 | 86.07 ± 0.72 | 42.85 ± 1.50 | 81.35 ± 3.31 | 23.17 ± 6.42 | 83.26 ± 0.44 | 19.77 ± 4.45 | **89.13 ± 1.81** | 49.39 ± 1.54 |
| ciprofloxacin | 80.24 ± 0.39 | 54.97 ± 1.18 | 79.77 ± 1.15 | 54.18 ± 3.12 | 82.21 ± 0.33 | 59.35 ± 0.62 | 80.76 ± 0.42 | 58.02 ± 0.79 | 83.50 ± 0.33 | 60.83 ± 0.59 | 85.44 ± 0.24 | 64.16 ± 0.46 | 82.67 ± 0.24 | 59.71 ± 1.52 | 84.46 ± 2.21 | 58.98 ± 0.86 | **86.09 ± 0.29** | **66.14 ± 1.16** |
| clindamycin | 81.65 ± 2.09 | 22.58 ± 1.27 | 84.21 ± 1.56 | 28.39 ± 7.35 | 86.49 ± 0.60 | 38.99 ± 5.93 | 82.93 ± 5.68 | 28.63 ± 10.49 | 84.51 ± 1.80 | 33.87 ± 2.48 | 83.57 ± 1.45 | 29.92 ± 4.37 | 86.92 ± 1.80 | 35.10 ± 7.06 | 71.59 ± 4.39 | 28.21 ± 3.30 | **89.88 ± 0.67** | 50.23 ± 7.67 |
| colistin | 64.04 ± 8.15 | 17.71 ± 3.46 | 72.32 ± 6.37 | 19.23 ± 3.82 | 69.83 ± 15.10 | 21.46 ± 9.81 | 77.53 ± 8.36 | 28.63 ± 10.49 | 66.23 ± 8.95 | 19.00 ± 3.41 | 61.74 ± 6.24 | 14.10 ± 1.66 | 75.62 ± 2.60 | **28.65 ± 2.15** | 78.02 ± 7.76 | 19.46 ± 2.25 | 68.04 ± 7.94 | 21.82 ± 11.69 |
| doripenem | 71.13 ± 5.41 | 32.40 ± 1.20 | 75.90 ± 7.91 | 75.69 ± 9.47 | 78.05 ± 6.21 | 79.04 ± 7.76 | 89.47 ± 4.60 | 62.14 ± 7.46 | 82.87 ± 5.17 | 83.73 ± 5.64 | 83.77 ± 5.16 | 83.79 ± 5.69 | **84.97 ± 7.22** | **85.05 ± 8.64** | 80.10 ± 2.37 | 78.97 ± 7.51 | 82.77 ± 10.78 | 83.83 ± 10.40 |
| ertapenem | 82.31 ± 0.82 | 40.77 ± 4.66 | 84.36 ± 1.08 | 34.89 ± 1.16 | 90.39 ± 1.18 | 45.48 ± 1.12 | 85.32 ± 0.94 | 58.64 ± 3.00 | 85.85 ± 1.62 | 53.35 ± 8.86 | 86.21 ± 1.13 | 53.82 ± 3.27 | **93.60 ± 2.55** | **75.01 ± 4.84** | 89.92 ± 2.10 | 42.57 ± 6.62 | 93.02 ± 0.44 | 64.03 ± 11.14 |
| erythromycin | 87.71 ± 1.42 | 47.47 ± 2.30 | 81.20 ± 2.05 | 46.15 ± 4.86 | 90.88 ± 1.27 | 58.28 ± 4.82 | 81.30 ± 1.40 | 60.77 ± 1.43 | 89.90 ± 0.66 | 50.89 ± 6.47 | 89.42 ± 0.88 | 50.56 ± 3.86 | 91.50 ± 0.46 | 54.53 ± 7.56 | 89.75 ± 3.03 | 53.04 ± 6.21 | **93.00 ± 0.46** | **65.02 ± 1.94** |
| gentamicin | 82.55 ± 1.06 | 64.56 ± 2.70 | 71.85 ± 2.21 | 61.91 ± 2.71 | 86.73 ± 3.97 | 59.67 ± 3.95 | 81.30 ± 0.46 | 27.77 ± 4.10 | 86.51 ± 0.88 | 59.49 ± 4.97 | 87.52 ± 0.70 | 61.93 ± 1.60 | 85.36 ± 1.28 | 57.71 ± 1.84 | 85.81 ± 1.06 | 59.44 ± 2.87 | 87.69 ± 0.94 | 63.23 ± 4.87 |
| imipenem | 90.92 ± 0.62 | 47.47 ± 5.07 | 89.20 ± 2.02 | 23.02 ± 1.92 | 77.19 ± 4.45 | 29.35 ± 6.66 | 77.35 ± 4.34 | 28.80 ± 2.41 | 93.70 ± 0.82 | 79.74 ± 1.28 | 93.12 ± 0.74 | 58.37 ± 1.65 | 94.18 ± 0.13 | 79.33 ± 1.22 | 77.24 ± 1.98 | 64.90 ± 6.61 | **94.53 ± 0.59** | **80.69 ± 2.68** |
| kanamycin | 64.70 ± 1.04 | 56.74 ± 1.30 | 71.85 ± 0.26 | 66.24 ± 1.12 | 79.21 ± 0.48 | 69.32 ± 1.70 | 78.34 ± 0.69 | 69.04 ± 1.25 | 78.18 ± 0.75 | 28.80 ± 3.49 | 77.84 ± 0.70 | 29.09 ± 2.92 | 77.24 ± 1.98 | 22.27 ± 1.98 | 79.97 ± 0.78 | 22.27 ± 1.98 | **81.34 ± 2.08** | 37.84 ± 2.78 |
| levofloxacin | 68.37 ± 1.84 | 22.59 ± 0.66 | 73.88 ± 1.64 | 21.57 ± 0.83 | 77.79 ± 1.16 | 57.97 ± 1.29 | 75.44 ± 0.59 | 47.88 ± 3.36 | 82.73 ± 0.99 | 66.67 ± 1.34 | 86.34 ± 0.45 | 79.94 ± 0.90 | 77.27 ± 0.94 | 71.42 ± 1.05 | 92.58 ± 1.49 | 68.14 ± 1.05 | 86.08 ± 0.52 | **81.91 ± 0.79** |
| meropenem | 94.12 ± 0.88 | 59.04 ± 3.95 | 93.73 ± 0.70 | 46.05 ± 2.29 | 95.97 ± 0.26 | 57.97 ± 2.45 | 96.43 ± 0.82 | 57.75 ± 7.88 | 96.43 ± 0.82 | 57.75 ± 7.88 | 79.03 ± 0.71 | 39.48 ± 2.36 | 75.83 ± 1.91 | 24.77 ± 3.19 | 75.88 ± 1.51 | 48.79 ± 3.97 | 84.75 ± 2.03 | 69.43 ± 0.88 |
| nalidixic acid | 73.70 ± 0.67 | 51.20 ± 10.97 | 72.87 ± 0.91 | 21.57 ± 0.83 | 77.79 ± 1.16 | 27.59 ± 2.45 | 98.15 ± 0.43 | 98.69 ± 0.25 | 98.55 ± 2.27 | 30.73 ± 3.25 | 97.22 ± 1.35 | 98.24 ± 0.83 | 96.54 ± 1.30 | 97.97 ± 0.77 | 96.97 ± 0.60 | 21.49 ± 0.62 | **96.92 ± 0.26** | 42.70 ± 4.50 |
| nitrofurantoin | 90.48 ± 8.35 | 83.31 ± 0.29 | 95.00 ± 4.59 | 94.66 ± 0.90 | 97.65 ± 0.84 | 98.47 ± 0.42 | 98.69 ± 0.25 | 98.69 ± 0.25 | 97.31 ± 1.16 | 98.24 ± 0.74 | 90.37 ± 6.06 | 60.87 ± 9.43 | 91.11 ± 0.36 | 64.75 ± 9.49 | 78.32 ± 3.87 | **97.86 ± 0.46** | **97.24 ± 0.40** | **98.19 ± 0.25** |
| oxacillin | 76.85 ± 12.08 | 47.15 ± 2.94 | 85.42 ± 1.35 | 62.42 ± 3.28 | 85.46 ± 6.33 | 61.36 ± 9.53 | 90.01 ± 5.05 | **98.69 ± 0.25** | 84.98 ± 9.48 | 68.11 ± 16.77 | 93.81 ± 0.63 | 96.31 ± 0.61 | 89.44 ± 1.14 | 74.75 ± 7.06 | 92.72 ± 0.91 | 79.24 ± 0.40 | 85.68 ± 3.99 | 69.21 ± 5.75 |
| penicillin | 64.09 ± 25.33 | 60.00 ± 0.96 | 91.37 ± 2.23 | 94.66 ± 0.90 | 90.12 ± 0.54 | 93.33 ± 1.10 | 94.31 ± 2.83 | **96.37 ± 2.02** | 93.41 ± 0.64 | 95.93 ± 0.52 | 96.31 ± 0.61 | 96.31 ± 0.36 | 89.44 ± 1.14 | 67.28 ± 1.40 | 84.28 ± 0.72 | 95.26 ± 0.43 | 92.30 ± 0.88 | **94.90 ± 1.00** |
| piperacillin-tazobactam | 81.84 ± 0.34 | 47.15 ± 2.94 | 80.82 ± 0.65 | 46.56 ± 1.45 | 89.49 ± 1.28 | 68.76 ± 6.35 | 87.84 ± 1.12 | 64.93 ± 5.61 | 89.66 ± 0.46 | 88.76 ± 6.35 | 83.54 ± 0.79 | 59.66 ± 4.99 | 78.41 ± 0.44 | 71.98 ± 5.26 | 76.70 ± 1.30 | 52.25 ± 2.42 | **90.44 ± 0.84** | 74.19 ± 4.78 |
| streptomycin | 69.88 ± 0.98 | 60.00 ± 0.96 | 72.75 ± 2.39 | 57.81 ± 2.31 | 79.39 ± 1.32 | 65.04 ± 1.75 | 80.19 ± 0.57 | 68.35 ± 1.42 | 84.52 ± 1.05 | 75.78 ± 0.91 | 76.43 ± 2.27 | 73.32 ± 1.37 | 87.77 ± 0.42 | 78.33 ± 1.46 | 75.02 ± 1.85 | 63.86 ± 1.89 | **84.77 ± 0.91** | **76.28 ± 1.28** |
| sulfamethoxazole | 67.20 ± 1.91 | 57.04 ± 0.04 | 75.68 ± 3.25 | 63.45 ± 2.46 | 77.28 ± 3.16 | 62.64 ± 3.89 | 77.65 ± 2.04 | 63.08 ± 2.93 | 76.11 ± 1.62 | 59.78 ± 1.51 | **90.93 ± 0.50** | **84.73 ± 0.46** | 82.80 ± 4.72 | 61.74 ± 2.28 | 81.49 ± 0.76 | 59.01 ± 2.08 | 76.15 ± 0.63 | 61.17 ± 2.21 |
| sulfisoxazole | 70.34 ± 0.46 | 56.74 ± 0.94 | 80.33 ± 1.62 | 62.12 ± 4.19 | 88.55 ± 0.77 | 79.66 ± 0.63 | 87.00 ± 1.67 | 73.56 ± 3.08 | 90.68 ± 0.73 | 83.47 ± 1.31 | 76.61 ± 11.40 | 16.08 ± 3.25 | 77.77 ± 2.28 | 62.20 ± 1.61 | 75.02 ± 1.85 | 62.20 ± 1.61 | **81.23 ± 0.66** | 61.17 ± 2.21 |
| teithromycin | 73.42 ± 2.11 | 9.77 ± 4.96 | 82.10 ± 3.27 | 9.29 ± 5.93 | 82.09 ± 2.66 | 12.04 ± 3.07 | 81.87 ± 3.54 | 10.31 ± 3.10 | 79.68 ± 5.11 | 15.38 ± 7.77 | 80.07 ± 0.30 | 76.28 ± 0.25 | 82.80 ± 4.72 | 17.26 ± 5.73 | 79.80 ± 5.65 | 6.55 ± 2.37 | **84.43 ± 5.78** | **25.38 ± 12.19** |
| tetracycline | 59.30 ± 0.58 | 58.20 ± 0.67 | 64.08 ± 0.15 | 60.32 ± 0.87 | 71.53 ± 0.56 | 67.32 ± 0.63 | 71.60 ± 0.86 | 67.20 ± 0.95 | 75.71 ± 0.57 | 71.43 ± 0.51 | 74.41 ± 1.41 | 71.43 ± 0.51 | 74.96 ± 0.50 | 70.75 ± 0.47 | 74.96 ± 0.50 | 70.75 ± 0.50 | **81.71 ± 0.34** | **79.07 ± 0.23** |
| tobramycin | 82.21 ± 2.01 | 55.51 ± 4.28 | 83.01 ± 0.66 | 57.92 ± 4.03 | 87.93 ± 1.51 | 69.18 ± 5.10 | 86.83 ± 1.24 | 67.77 ± 3.40 | 86.69 ± 3.28 | 67.15 ± 8.62 | 86.84 ± 2.12 | 67.68 ± 6.40 | 87.73 ± 0.70 | 71.30 ± 3.07 | 84.06 ± 1.39 | 70.75 ± 0.47 | **89.92 ± 0.60** | 76.53 ± 1.78 |
| trimethoprim-sulfamethoxazole | 85.78 ± 0.58 | 47.47 ± 0.92 | 82.55 ± 0.28 | 46.81 ± 2.77 | 88.88 ± 0.33 | 64.36 ± 1.46 | 87.65 ± 0.51 | 62.72 ± 1.04 | 91.08 ± 0.53 | 69.30 ± 1.01 | **91.61 ± 0.86** | 71.11 ± 2.02 | 87.18 ± 0.07 | 59.38 ± 1.54 | 88.95 ± 1.00 | 63.99 ± 0.44 | 90.52 ± 0.57 | 70.13 ± 1.72 |

Supplementary Table 14: Per-antibiotic MIC regression prediction results across 3 seeds. Values are mean ± std. Bold indicates the best mean for each metric.

| Antibiotic | Mistral-DNA Pearson | Mistral-DNA R² | DNABERT-2 Pearson | DNABERT-2 R² | Nucleotide Transformer Pearson | Nucleotide Transformer R² | ProkBERT Pearson | ProkBERT R² | ESM-2 Pearson | ESM-2 R² | ESM-C Pearson | ESM-C R² | ProtBERT Pearson | ProtBERT R² | gLM2 Pearson | gLM2 R² | Bacformer Pearson | Bacformer R² |
|---|---|---|---|---|---|---|---|---|---|---|---|---|---|---|---|---|---|---|
| amikacin | 67.35 ± 0.91 | 45.08 ± 1.48 | 68.37 ± 1.03 | 46.63 ± 1.44 | 70.01 ± 1.02 | 48.82 ± 1.47 | 69.78 ± 0.74 | 48.66 ± 1.04 | 69.95 ± 1.08 | 48.63 ± 1.71 | 71.13 ± 0.99 | **50.50 ± 1.47** | 69.36 ± 0.85 | 48.01 ± 1.21 | 69.42 ± 1.67 | 48.00 ± 2.42 | **71.32 ± 1.59** | 50.46 ± 2.08 |
| amoxicillin-clavulanic acid | 46.27 ± 1.45 | 21.06 ± 1.03 | 48.59 ± 2.53 | 23.24 ± 2.06 | 51.50 ± 4.26 | 26.41 ± 4.21 | 52.04 ± 1.26 | 26.89 ± 1.14 | 53.13 ± 0.81 | 27.55 ± 0.93 | 56.14 ± 0.48 | 30.95 ± 0.30 | 54.47 ± 0.93 | 29.13 ± 1.12 | 51.14 ± 1.11 | 25.24 ± 1.01 | **57.28 ± 0.47** | **32.09 ± 0.70** |
| ampicillin | 46.43 ± 0.78 | 20.94 ± 0.81 | 51.01 ± 0.68 | 25.35 ± 1.03 | 52.72 ± 0.51 | 27.40 ± 0.87 | 51.78 ± 0.63 | 26.68 ± 0.69 | 54.19 ± 0.75 | 28.81 ± 0.92 | **57.00 ± 0.24** | **31.94 ± 0.41** | 52.20 ± 1.74 | 26.87 ± 1.74 | 51.62 ± 0.07 | 25.74 ± 0.48 | 57.06 ± 1.09 | 31.79 ± 1.36 |
| ampicillin-subactam | 40.77 ± 4.28 | 16.57 ± 2.74 | 41.23 ± 3.29 | 16.77 ± 2.37 | 45.36 ± 5.54 | 20.43 ± 4.64 | 42.14 ± 3.78 | 17.23 ± 2.85 | 45.49 ± 2.52 | 20.52 ± 2.23 | 46.60 ± 3.46 | 21.33 ± 2.96 | 42.40 ± 1.66 | 17.46 ± 1.20 | 47.21 ± 3.75 | 22.04 ± 3.45 | **47.78 ± 7.02** | **22.09 ± 7.02** |
| azithromycin | 79.15 ± 0.38 | 62.59 ± 0.66 | 79.57 ± 0.06 | 63.18 ± 0.10 | 80.01 ± 0.16 | 63.94 ± 0.21 | 79.74 ± 0.05 | 63.42 ± 0.09 | 79.63 ± 0.07 | 63.31 ± 0.15 | **80.43 ± 0.58** | **64.50 ± 1.02** | 79.97 ± 0.14 | 63.82 ± 0.24 | 79.73 ± 0.14 | 63.40 ± 0.28 | 80.27 ± 0.33 | 64.33 ± 0.70 |
| aztreonam | 72.00 ± 1.08 | 50.51 ± 2.22 | 72.36 ± 1.57 | 51.22 ± 2.83 | 74.70 ± 2.30 | 55.04 ± 3.92 | 74.48 ± 2.44 | 54.14 ± 4.96 | 77.48 ± 1.57 | 59.54 ± 2.81 | 78.07 ± 2.12 | 60.23 ± 3.90 | 77.07 ± 2.08 | 58.61 ± 3.02 | 74.41 ± 0.82 | 54.77 ± 1.17 | **79.38 ± 1.96** | **62.61 ± 3.12** |
| cefazolin | 65.50 ± 0.57 | 41.05 ± 1.29 | 66.18 ± 1.10 | 43.27 ± 1.77 | 69.40 ± 1.10 | 47.84 ± 1.98 | 68.16 ± 1.44 | 46.19 ± 2.16 | 71.38 ± 1.29 | 50.22 ± 2.12 | **75.37 ± 1.85** | **56.05 ± 2.89** | 67.57 ± 1.06 | 45.26 ± 1.74 | 70.19 ± 0.19 | 48.77 ± 0.96 | 73.18 ± 0.83 | 53.21 ± 1.30 |
| cefepime | 48.40 ± 2.99 | 23.05 ± 2.62 | 49.95 ± 2.10 | 24.28 ± 1.43 | 53.21 ± 2.92 | 28.21 ± 3.25 | 53.28 ± 1.58 | 28.14 ± 1.85 | 58.39 ± 3.10 | 33.82 ± 3.44 | 59.39 ± 1.85 | 34.51 ± 2.59 | 56.44 ± 3.72 | 31.56 ± 4.15 | 52.34 ± 0.59 | 27.11 ± 0.72 | **60.53 ± 1.95** | **36.34 ± 2.49** |
| cefotaxime | 9.42 ± 6.07 | -2.14 ± 5.56 | 16.40 ± 8.74 | 3.07 ± 4.55 | 51.74 ± 9.12 | 25.09 ± 9.05 | 49.80 ± 1.62 | 12.55 ± 2.54 | 57.33 ± 1.90 | 31.67 ± 2.40 | 58.41 ± 2.50 | 33.13 ± 3.77 | 54.12 ± 3.05 | 27.74 ± 3.29 | 46.18 ± 4.42 | 20.12 ± 5.28 | **63.09 ± 1.48** | **38.04 ± 3.43** |
| cefoxitin | 54.85 ± 1.67 | 29.82 ± 1.56 | 60.37 ± 0.97 | 36.04 ± 0.97 | 63.07 ± 0.88 | 39.67 ± 1.15 | 63.50 ± 0.88 | 39.75 ± 1.05 | 62.75 ± 1.06 | 39.14 ± 1.18 | 63.51 ± 1.20 | 39.75 ± 1.20 | 63.16 ± 0.61 | 39.75 ± 0.69 | 61.36 ± 0.39 | 37.35 ± 0.50 | **66.00 ± 1.27** | **43.21 ± 1.46** |
| ceftazidime | 32.75 ± 4.16 | 10.27 ± 2.16 | 32.72 ± 3.35 | 10.50 ± 1.46 | 40.53 ± 4.54 | 15.11 ± 2.73 | 42.72 ± 1.25 | 16.85 ± 0.90 | 47.28 ± 0.40 | 21.11 ± 0.57 | 48.70 ± 0.38 | 22.66 ± 1.90 | 50.81 ± 1.20 | 25.48 ± 1.95 | 49.58 ± 1.50 | 24.17 ± 1.82 | **52.51 ± 1.19** | **26.55 ± 1.40** |
| ceftriaxone | 44.76 ± 0.70 | 19.70 ± 0.71 | 46.81 ± 0.74 | 21.63 ± 0.69 | 54.96 ± 1.97 | 29.66 ± 1.73 | 54.68 ± 1.50 | 29.38 ± 1.29 | 60.87 ± 0.62 | 37.00 ± 0.81 | 60.87 ± 1.20 | 36.49 ± 0.84 | 50.81 ± 1.20 | 25.48 ± 1.95 | 49.58 ± 1.50 | 24.17 ± 1.82 | 61.88 ± 1.62 | 36.95 ± 2.55 |
| cephalexin | 4.56 ± 1.02 | -0.02 ± 0.30 | 38.41 ± 1.93 | 9.77 ± 0.87 | 37.17 ± 9.23 | 11.15 ± 6.00 | 38.19 ± 2.72 | 12.09 ± 1.73 | 37.37 ± 0.13 | 20.69 ± 0.67 | 41.84 ± 1.81 | 16.04 ± 2.22 | 41.84 ± 1.81 | 16.04 ± 2.22 | 32.45 ± 3.89 | 9.84 ± 2.07 | 45.78 ± 2.60 | 19.01 ± 2.91 |
| cephalothin | 56.54 ± 0.78 | 31.78 ± 1.01 | 57.93 ± 1.22 | 33.37 ± 1.43 | 63.17 ± 1.83 | 39.66 ± 2.12 | 62.08 ± 1.43 | 38.37 ± 1.64 | 64.81 ± 1.96 | 41.65 ± 2.45 | **69.00 ± 1.59** | **47.11 ± 2.05** | 63.53 ± 1.58 | 39.97 ± 1.81 | 62.63 ± 1.41 | 38.71 ± 1.61 | 67.66 ± 1.71 | 45.43 ± 2.16 |
| chloramphenicol | 6.48 ± 2.58 | 0.05 ± 0.19 | 7.01 ± 4.62 | 0.24 ± 0.54 | 19.53 ± 6.34 | 3.33 ± 1.91 | 18.82 ± 3.22 | 2.25 ± 1.92 | 16.96 ± 4.27 | 2.38 ± 2.25 | 35.04 ± 5.97 | 9.38 ± 2.38 | 18.65 ± 4.38 | 2.94 ± 0.97 | 15.51 ± 13.44 | 2.38 ± 3.24 | **35.36 ± 12.54** | **10.53 ± 7.83** |
| ciprofloxacin | 58.16 ± 8.99 | 30.88 ± 10.42 | 58.18 ± 8.67 | 30.84 ± 9.15 | 59.02 ± 8.53 | 31.58 ± 11.04 | 63.20 ± 8.52 | 37.95 ± 11.09 | 61.98 ± 8.07 | 37.00 ± 10.81 | 60.87 ± 8.15 | 35.29 ± 10.44 | 63.70 ± 10.05 | 36.95 ± 14.57 | 60.98 ± 7.69 | 35.71 ± 10.05 | **66.14 ± 10.27** | **41.47 ± 13.32** |
| clindamycin | 57.09 ± 0.62 | 1.27 ± 0.12 | 16.99 ± 1.51 | 2.82 ± 0.44 | 58.82 ± 0.82 | 34.12 ± 1.36 | 58.32 ± 0.40 | 33.72 ± 0.76 | 29.32 ± 2.20 | 7.60 ± 0.94 | 26.42 ± 5.41 | 4.62 ± 4.21 | 58.53 ± 1.48 | 34.02 ± 1.55 | 19.72 ± 4.39 | 3.64 ± 1.38 | 10.99 ± 0.56 | 10.99 ± 0.56 |
| colistin | 55.41 ± 2.48 | 31.93 ± 0.61 | 57.20 ± 3.11 | 32.40 ± 1.58 | 58.82 ± 3.22 | 34.12 ± 1.36 | 64.95 ± 4.05 | 41.74 ± 0.79 | 59.10 ± 0.89 | 34.43 ± 0.89 | 59.13 ± 1.11 | 34.60 ± 1.06 | 58.53 ± 1.48 | 34.02 ± 1.55 | 56.44 ± 0.48 | 3.64 ± 1.38 | 61.75 ± 0.67 | 34.47 ± 1.39 |
| daptomycin | 22.28 ± 6.30 | 4.73 ± 2.59 | 26.95 ± 3.60 | 7.03 ± 1.72 | 61.61 ± 4.06 | 16.64 ± 3.73 | 40.77 ± 3.28 | 15.81 ± 1.90 | 32.42 ± 3.91 | 9.81 ± 1.90 | 43.88 ± 6.65 | 17.47 ± 4.24 | 45.48 ± 5.02 | 19.00 ± 4.77 | 32.54 ± 4.17 | 8.86 ± 1.53 | 50.27 ± 7.21 | 49.06 ± 4.04 |
| doripenem | 39.30 ± 20.02 | 13.52 ± 13.63 | 40.87 ± 18.10 | 13.90 ± 11.44 | 46.90 ± 9.64 | 21.12 ± 10.00 | 52.22 ± 4.53 | 25.08 ± 4.64 | 43.66 ± 15.77 | 15.37 ± 18.13 | 46.33 ± 14.40 | 11.59 ± 30.18 | 43.99 ± 15.22 | 13.23 ± 22.82 | 54.45 ± 4.38 | 26.53 ± 5.14 | 53.99 ± 9.93 | 22.54 ± 5.30 |
| doxycycline | 44.81 ± 7.03 | 18.72 ± 6.94 | 44.48 ± 7.11 | 19.01 ± 6.14 | 50.00 ± 4.35 | 23.16 ± 4.30 | 45.99 ± 4.07 | 20.23 ± 2.64 | 47.62 ± 13.69 | 22.04 ± 14.10 | 60.01 ± 6.35 | 33.93 ± 6.73 | 47.19 ± 7.18 | 18.55 ± 17.14 | 48.11 ± 5.74 | 22.84 ± 5.62 | 51.85 ± 3.34 | 28.48 ± 9.28 |
| enrofloxacin | 48.62 ± 2.95 | 23.17 ± 3.47 | 50.81 ± 2.43 | 25.53 ± 2.70 | 52.66 ± 1.16 | 28.20 ± 2.01 | 53.69 ± 1.54 | 28.20 ± 2.02 | 57.48 ± 1.18 | 32.22 ± 1.93 | 57.84 ± 1.50 | 32.58 ± 1.63 | 53.76 ± 1.33 | 28.59 ± 1.63 | 51.09 ± 2.47 | 26.02 ± 2.57 | 58.22 ± 1.07 | 25.74 ± 2.71 |
| ertapenem | 23.72 ± 2.70 | 5.16 ± 0.99 | 26.65 ± 1.42 | 6.82 ± 0.52 | 47.42 ± 2.82 | 20.99 ± 2.01 | 51.24 ± 2.65 | 23.83 ± 2.21 | 58.55 ± 1.40 | 32.80 ± 1.40 | 62.78 ± 1.30 | 38.78 ± 1.71 | 57.04 ± 3.52 | 20.45 ± 3.40 | 39.53 ± 1.28 | 14.29 ± 1.70 | 60.94 ± 3.94 | 33.35 ± 4.48 |
| erythromycin | 64.83 ± 1.13 | 40.73 ± 2.02 | 66.07 ± 0.89 | 42.60 ± 1.09 | 53.06 ± 0.29 | 26.90 ± 0.36 | 54.88 ± 2.60 | 28.60 ± 1.17 | 49.68 ± 1.48 | 29.53 ± 1.77 | 69.96 ± 1.36 | 47.94 ± 1.93 | 54.07 ± 2.06 | 27.51 ± 1.54 | 39.23 ± 2.09 | 39.23 ± 2.09 | 71.99 ± 1.53 | 51.63 ± 2.18 |
| florfenicol | 42.05 ± 1.24 | 16.50 ± 0.80 | 50.18 ± 1.28 | 23.97 ± 1.66 | 59.19 ± 1.02 | 26.90 ± 0.36 | 58.66 ± 0.67 | 28.60 ± 2.60 | 56.35 ± 2.57 | 25.19 ± 0.31 | 54.07 ± 2.06 | 24.14 ± 7.30 | 54.07 ± 2.06 | 27.51 ± 1.54 | 52.03 ± 2.22 | 25.13 ± 0.98 | 59.05 ± 1.36 | 33.53 ± 0.57 |
| gentamicin | 55.79 ± 0.61 | 30.47 ± 0.75 | 58.11 ± 0.79 | 33.46 ± 0.94 | 59.19 ± 1.02 | 34.76 ± 1.43 | 58.66 ± 0.67 | 34.25 ± 0.83 | 59.58 ± 0.22 | 35.19 ± 0.31 | 58.61 ± 0.60 | 34.24 ± 0.68 | 58.71 ± 1.17 | 34.30 ± 1.36 | 58.71 ± 1.17 | 34.30 ± 1.36 | 60.23 ± 0.83 | 36.01 ± 0.92 |
| imipenem | 70.39 ± 1.64 | 49.28 ± 2.42 | 70.87 ± 1.50 | 49.89 ± 2.37 | 71.28 ± 1.70 | 50.45 ± 2.60 | 66.82 ± 0.48 | 18.89 ± 7.40 | 71.83 ± 1.81 | 51.33 ± 2.78 | 71.61 ± 1.39 | 51.62 ± 1.96 | 72.22 ± 1.30 | 51.62 ± 2.30 | 72.16 ± 0.94 | 50.42 ± 1.38 | 73.39 ± 1.88 | 53.62 ± 2.66 |
| kanamycin | 0.75 ± 5.27 | -0.19 ± 0.24 | 23.16 ± 2.69 | 4.62 ± 0.85 | 36.53 ± 2.17 | 12.36 ± 1.32 | 58.66 ± 2.40 | 12.97 ± 1.59 | 34.85 ± 4.49 | 11.67 ± 2.87 | 73.68 ± 8.23 | 11.61 ± 2.32 | 72.22 ± 2.35 | 35.36 ± 2.35 | 30.12 ± 2.79 | 8.57 ± 1.94 | 58.55 ± 1.18 | 12.39 ± 2.59 |
| levofloxacin | 28.22 ± 10.03 | 7.61 ± 4.71 | 42.69 ± 14.73 | 17.54 ± 5.91 | 36.53 ± 2.17 | 17.75 ± 6.00 | 46.61 ± 6.15 | 18.89 ± 2.40 | 35.69 ± 2.59 | 11.67 ± 2.87 | 49.40 ± 4.47 | 32.76 ± 2.13 | 35.36 ± 2.35 | 11.61 ± 2.80 | 32.95 ± 1.47 | 24.93 ± 1.49 | 33.45 ± 1.54 | 33.45 ± 1.54 |
| marbofloxacin | 8.05 ± 15.49 | 0.22 ± 2.85 | 7.73 ± 14.73 | -5.96 ± 10.47 | 5.89 ± 2.76 | 12.96 ± 1.32 | 16.07 ± 17.49 | -7.48 ± 20.22 | 10.06 ± 18.14 | -5.13 ± 12.04 | 3.85 ± 25.12 | -14.01 ± 26.52 | 49.40 ± 4.47 | -21.28 ± 35.29 | -15.08 ± 25.90 | 2.86 ± 9.77 | -34.65 ± 30.70 | 30.70 ± 8.55 |
| meropenem | 41.18 ± 6.12 | 15.70 ± 6.36 | 42.54 ± 4.39 | 16.75 ± 5.53 | 50.89 ± 2.76 | 24.93 ± 3.20 | 49.17 ± 10.47 | 23.46 ± 11.50 | 61.45 ± 4.60 | 35.53 ± 5.81 | 65.92 ± 2.60 | 42.74 ± 4.01 | 53.88 ± 4.92 | 28.06 ± 5.22 | 47.65 ± 9.41 | 21.83 ± 5.45 | 37.09 ± 8.55 | 37.09 ± 8.55 |
| minocycline | 73.25 ± 2.30 | 52.37 ± 2.49 | 74.71 ± 0.93 | 55.22 ± 0.88 | 76.01 ± 1.55 | 57.41 ± 2.41 | 76.01 ± 1.55 | 37.25 ± 3.48 | 75.50 ± 0.90 | 56.73 ± 1.26 | 75.93 ± 1.05 | 57.42 ± 1.49 | 76.12 ± 1.23 | 53.49 ± 1.07 | 73.45 ± 0.80 | 53.49 ± 1.07 | 76.95 ± 1.48 | 58.78 ± 1.98 |
| nalidixic acid | 51.40 ± 1.91 | 25.51 ± 1.27 | 59.32 ± 0.72 | 33.72 ± 0.48 | 61.16 ± 2.64 | 36.28 ± 2.54 | 61.22 ± 3.01 | 37.25 ± 3.48 | 69.82 ± 1.51 | 47.41 ± 1.35 | 70.19 ± 1.09 | 47.42 ± 2.01 | 65.91 ± 1.33 | 42.53 ± 1.68 | 60.95 ± 1.28 | 36.28 ± 1.27 | 66.34 ± 4.26 | 43.54 ± 5.42 |
| nitrofurantoin | 42.25 ± 2.47 | 17.63 ± 1.76 | 43.20 ± 2.09 | 18.45 ± 1.44 | 45.64 ± 2.06 | 20.62 ± 1.96 | 43.62 ± 2.13 | 19.01 ± 1.85 | 47.84 ± 1.24 | 22.59 ± 1.14 | 49.79 ± 1.83 | 24.44 ± 1.51 | 45.39 ± 1.63 | 22.08 ± 1.14 | 46.65 ± 2.80 | 19.76 ± 2.41 | 54.10 ± 1.14 | 28.32 ± 1.20 |
| orbifloxacin | 91.48 ± 1.91 | 62.92 ± 2.32 | 94.36 ± 0.93 | 88.58 ± 1.68 | 94.72 ± 1.12 | 89.60 ± 2.22 | 95.05 ± 1.36 | 90.30 ± 2.51 | 94.93 ± 0.94 | 90.05 ± 1.78 | 94.87 ± 0.81 | 89.97 ± 1.50 | 95.10 ± 1.00 | 90.42 ± 1.88 | 94.76 ± 1.08 | 89.78 ± 2.02 | 95.18 ± 1.05 | 90.55 ± 1.99 |
| oxacillin | 6.61 ± 4.56 | -4.38 ± 4.73 | 10.49 ± 5.31 | -2.52 ± 3.35 | 10.79 ± 1.15 | -5.91 ± 6.96 | 3.74 ± 1.34 | -5.61 ± 5.73 | 14.22 ± 8.99 | -4.05 ± 4.23 | 24.15 ± 9.09 | -2.85 ± 13.33 | 8.81 ± 7.84 | -5.34 ± 5.05 | 17.65 ± 9.42 | -0.58 ± 3.66 | 26.94 ± 13.87 | 4.06 ± 3.81 |
| penicillin | 39.98 ± 5.97 | 13.98 ± 6.21 | 54.68 ± 3.43 | 29.23 ± 3.52 | 56.74 ± 2.49 | 30.98 ± 2.18 | 63.44 ± 0.42 | 38.44 ± 0.91 | 64.06 ± 1.88 | 40.62 ± 2.06 | 66.14 ± 3.02 | 43.00 ± 3.14 | 62.72 ± 3.66 | 38.87 ± 4.20 | 58.80 ± 4.05 | 33.19 ± 3.74 | 70.36 ± 2.44 | 48.92 ± 2.99 |
| piperacillin-tazobactam | 39.90 ± 1.98 | 15.08 ± 1.72 | 43.53 ± 2.53 | 18.16 ± 3.47 | 48.33 ± 2.03 | 22.67 ± 2.41 | 49.51 ± 3.17 | 23.54 ± 3.36 | 55.46 ± 3.46 | 25.16 ± 5.06 | 55.16 ± 1.08 | 29.68 ± 5.60 | 52.99 ± 2.74 | 27.62 ± 3.34 | 49.39 ± 2.46 | 23.52 ± 3.01 | 60.02 ± 4.51 | 35.04 ± 5.81 |
| polymyxin B | 56.97 ± 0.92 | 32.01 ± 1.01 | 58.82 ± 0.85 | 34.18 ± 0.79 | 67.38 ± 2.28 | 44.81 ± 2.91 | 66.31 ± 1.20 | 40.41 ± 2.44 | 66.88 ± 2.18 | 67.07 ± 1.08 | 71.07 ± 1.08 | 49.57 ± 1.36 | 67.21 ± 0.99 | 44.58 ± 0.79 | 64.52 ± 0.58 | 23.52 ± 2.04 | 72.30 ± 2.04 | 51.81 ± 2.56 |
| pradofloxacin | 23.37 ± 1.83 | 0.06 ± 8.91 | 29.67 ± 9.17 | 6.46 ± 8.12 | 28.36 ± 4.31 | 18.56 ± 4.72 | 46.57 ± 3.09 | 20.41 ± 4.24 | 46.17 ± 6.72 | 18.82 ± 8.05 | 44.44 ± 11.11 | 19.31 ± 10.42 | 35.33 ± 2.14 | 21.02 ± 6.42 | 31.96 ± 13.30 | 8.46 ± 9.67 | 17.56 ± 9.45 | 17.56 ± 9.45 |
| rifampin | 13.71 ± 4.66 | 1.44 ± 0.97 | 14.28 ± 0.85 | 1.47 ± 0.25 | 46.45 ± 0.99 | 7.07 ± 3.19 | 35.05 ± 11.51 | 9.98 ± 5.43 | 44.88 ± 4.54 | 18.82 ± 4.54 | 48.88 ± 4.54 | 21.93 ± 3.73 | 35.33 ± 3.14 | 10.88 ± 3.04 | 33.54 ± 11.82 | 8.98 ± 4.66 | 51.13 ± 3.26 | 25.49 ± 2.82 |
| streptomycin | 65.70 ± 0.55 | 38.90 ± 12.54 | 65.43 ± 11.25 | 42.08 ± 13.97 | 46.45 ± 0.99 | 21.34 ± 0.77 | 67.21 ± 8.07 | 44.45 ± 1.87 | 68.89 ± 4.92 | 46.59 ± 11.82 | 66.77 ± 7.60 | 67.05 ± 8.26 | 62.28 ± 5.46 | 37.65 ± 6.19 | 67.05 ± 8.26 | 41.83 ± 12.62 | 58.45 ± 6.02 | 46.60 ± 8.24 |
| sulfamethoxazole | 34.52 ± 2.22 | 11.36 ± 1.24 | 41.54 ± 2.45 | 16.58 ± 1.95 | 46.45 ± 0.99 | 21.34 ± 0.77 | 45.58 ± 1.35 | 20.27 ± 0.78 | 55.58 ± 1.49 | 28.34 ± 9.00 | 54.86 ± 2.49 | 62.28 ± 2.49 | 41.41 ± 5.15 | 16.96 ± 3.95 | 43.94 ± 1.97 | 44.57 ± 11.06 | 57.69 ± 2.42 | 32.43 ± 2.43 |
| sulfisoxazole | 19.04 ± 3.72 | -0.74 ± 1.05 | 11.66 ± 15.25 | -2.67 ± 2.93 | 18.94 ± 2.23 | 2.40 ± 1.94 | 20.07 ± 2.99 | 3.08 ± 2.02 | 22.66 ± 0.37 | 5.68 ± 5.18 | 27.12 ± 5.70 | 5.91 ± 2.99 | 10.19 ± 2.35 | -0.73 ± 1.21 | 24.86 ± 5.28 | 4.20 ± 3.41 | 23.63 ± 3.14 | 2.55 ± 2.83 |
| telithromycin | 19.83 ± 0.36 | 3.56 ± 0.64 | 45.59 ± 2.22 | 19.43 ± 1.51 | 55.93 ± 3.70 | 30.50 ± 4.43 | 53.53 ± 1.14 | 27.34 ± 1.24 | 62.66 ± 0.37 | 28.34 ± 9.00 | 65.77 ± 0.46 | 42.18 ± 0.92 | 57.25 ± 1.24 | 31.15 ± 1.75 | 46.49 ± 1.57 | 20.29 ± 1.30 | 60.71 ± 1.40 | 35.84 ± 1.63 |
| tetracycline | 41.16 ± 2.74 | 4.31 ± 2.49 | 33.08 ± 1.60 | 10.83 ± 1.00 | 42.63 ± 3.04 | 12.43 ± 2.23 | 41.81 ± 1.89 | 16.78 ± 1.28 | 45.30 ± 3.50 | 19.37 ± 3.01 | 45.50 ± 2.78 | 19.87 ± 2.55 | 42.61 ± 3.40 | 17.77 ± 3.03 | 45.46 ± 1.50 | 11.00 ± 1.42 | 60.27 ± 1.24 | 20.27 ± 0.62 |
| tigecycline | 25.72 ± 0.94 | 6.50 ± 0.38 | 48.96 ± 4.84 | 23.52 ± 4.10 | 36.88 ± 2.22 | 13.39 ± 1.55 | 39.37 ± 1.38 | 15.15 ± 0.91 | 44.68 ± 1.24 | 19.60 ± 0.79 | 49.46 ± 1.65 | 23.74 ± 1.49 | 40.02 ± 7.56 | 16.03 ± 5.47 | 45.46 ± 1.50 | 20.04 ± 1.46 | 52.35 ± 0.84 | 26.20 ± 1.24 |
| tobramycin | 37.97 ± 3.94 | 13.42 ± 2.80 | 52.86 ± 2.62 | 27.33 ± 2.02 | 54.22 ± 3.28 | 24.83 ± 3.49 | 51.92 ± 3.85 | 26.81 ± 1.90 | 55.00 ± 3.40 | 29.33 ± 3.28 | 54.76 ± 4.72 | 29.39 ± 4.57 | 53.01 ± 4.70 | 27.83 ± 5.41 | 51.21 ± 1.17 | 29.42 ± 2.32 | 59.21 ± 3.78 | 34.44 ± 4.14 |
| trimethoprim-sulfamethoxazole | 49.90 ± 1.32 | 24.02 ± 1.46 | 61.77 ± 7.48 | 37.34 ± 1.71 | 55.93 ± 4.02 | 28.83 ± 3.49 | 64.09 ± 1.22 | 40.80 ± 1.66 | 66.66 ± 0.86 | 33.17 | 55.15 ± 4.08 | 32.09 ± 3.30 | 57.85 ± 3.87 | 32.12 ± 5.04 | 63.59 ± 0.45 | 39.63 ± 0.64 | 59.92 ± 2.89 | 35.16 ± 2.48 |
| vancomycin | 14.32 ± 5.27 | -0.57 ± 4.11 | 30.27 ± 7.49 | 8.39 ± 4.89 | 49.51 ± 4.69 | 23.33 ± 5.10 | 43.46 ± 10.55 | 17.68 ± 10.86 | 39.25 ± 8.09 | 13.18 ± 6.24 | 51.69 ± 2.51 | 25.56 ± 2.71 | 40.21 ± 10.64 | 12.70 ± 10.03 | 33.83 ± 9.94 | 10.08 ± 8.51 | 45.52 ± 8.99 | 18.05 ± 10.09 |

## C.6 PHENOTYPIC TRAITS PREDICTION

We measure the AUPRC across phenotype groups (Supp. Fig. 13) as well as overall performance across metrics and all phenotypic traits (Supp. Table 15), and groups (Supp. Table 16). Finally, we include the results for each individual phenotype (Supp. Table 17 & 18). Analyzing performance across phenotype groups and metrics (Supp. Fig. 13 & Supp. Table 16), bLM-Bacformer achieves the best or combined best result across all groups and metrics, showing the benefits of incorporating genomic interactions into phenotype prediction models. Across metrics and phenotypes, Bacformer achieves the best result on 80 phenotypes on AUROC, 78 on AUPRC and 86 on F1 out of a total of 139 phenotypes. Therefore, showing that there is a considerable variation between phenotypes and one should choose a model specific for a phenotype. We believe this is due to the variable number of labels available for a phenotype as well as the inherent differences in the phenotypes themselves, which make some phenotypes easier to predict using a computational approach than others.

Predicting phenotypes is often a very challenging task which includes understanding the effect of mutations and multi-level genomic interactions. However, accurately predicting phenotypes could allow us to engineer genomes for a desired purpose, such as sustainable bioproduction, thus having potentially massive positive impact. We believe that next-generation models should consider the genomic context and incorporate the prior knowledge, such as genome-scale metabolic models to make the most out of available data for a given phenotype.

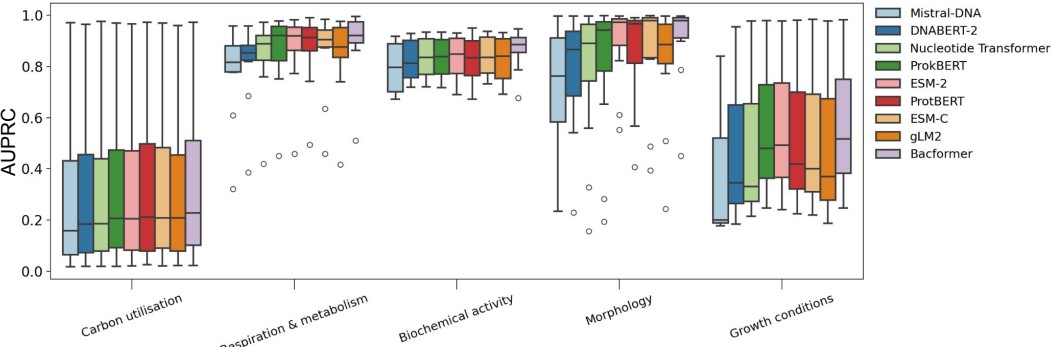

Supplementary Figure 13: AUPRC across diverse phenotypic traits groups and methods. The box spans the inter-quartile range with a line marking the median value. The results were macro-averaged across classes for each phenotype.

Supplementary Table 15: Overall phenotypic traits prediction performance across all phenotypes. Values are mean ± standard deviation across 5 seeds. The results were macro-averaged across classes for each phenotype.

| Method | AUROC | AUPRC | F1 |
|---|---|---|---|
| **Mistral-DNA** | $60.34 \pm 9.81$ | $39.72 \pm 33.09$ | $47.25 \pm 6.67$ |
| **DNABERT-2** | $63.83 \pm 11.06$ | $42.10 \pm 34.24$ | $49.42 \pm 8.99$ |
| **Nucleotide Transformer** | $65.98 \pm 11.45$ | $43.22 \pm 34.85$ | $52.04 \pm 11.07$ |
| **ProkBERT** | $67.64 \pm 11.87$ | $44.80 \pm 34.72$ | $52.03 \pm 11.56$ |
| **ESM-2** | $68.71 \pm 12.36$ | $45.61 \pm 35.45$ | $53.10 \pm 11.90$ |
| **ESM-C** | $69.07 \pm 11.98$ | $45.54 \pm 35.38$ | $52.28 \pm 11.39$ |
| **ProtBERT** | $68.47 \pm 11.97$ | $45.47 \pm 34.89$ | $53.71 \pm 11.65$ |
| **gLM2** | $65.82 \pm 11.41$ | $44.00 \pm 34.40$ | $51.11 \pm 10.86$ |
| **Bacformer** | $\mathbf{71.34 \pm 12.91}$ | $\mathbf{47.80 \pm 35.99}$ | $\mathbf{56.14 \pm 13.19}$ |

Supplementary Table 16: Phenotype-group prediction results; *mean ± standard deviation* over five random seeds. The highest score in each column is shown in bold. Results are macro-averaged across classes for each phenotype.

| Method | Biochemical activity | | | Carbon utilisation | | | Growth conditions | | | Morphology | | | Respiration & metabolism | | |
|---|---|---|---|---|---|---|---|---|---|---|---|---|---|---|---|
| | AUROC | AUPRC | F1 | AUROC | AUPRC | F1 | AUROC | AUPRC | F1 | AUROC | AUPRC | F1 | AUROC | AUPRC | F1 |
| Mistral-DNA | 54.67 ± 12.73 | 79.60 ± 10.06 | 44.48 ± 2.40 | 58.17 ± 7.25 | 27.95 ± 27.92 | 46.67 ± 3.25 | 63.73 ± 4.34 | 40.58 ± 37.70 | 36.80 ± 17.74 | 70.07 ± 11.96 | 72.05 ± 23.63 | 52.81 ± 9.92 | 71.79 ± 11.54 | 77.30 ± 18.47 | 50.88 ± 14.54 |
| DNABERT-2 | 59.88 ± 9.16 | 82.39 ± 8.55 | 46.19 ± 6.05 | 60.46 ± 7.60 | 29.24 ± 28.29 | 47.63 ± 3.90 | 71.22 ± 17.07 | 49.41 ± 40.69 | 45.89 ± 27.95 | 79.15 ± 12.39 | 79.75 ± 22.57 | 56.83 ± 14.23 | 76.88 ± 11.06 | 80.97 ± 16.72 | 60.52 ± 15.97 |
| Nucleotide Transformer | 63.74 ± 8.59 | 83.40 ± 8.47 | 50.62 ± 4.88 | 62.07 ± 7.58 | 30.06 ± 28.72 | 49.20 ± 4.03 | 79.33 ± 13.37 | 50.82 ± 41.12 | 51.73 ± 33.88 | 79.63 ± 14.10 | 78.94 ± 26.27 | 61.55 ± 20.66 | 81.74 ± 10.26 | 83.90 ± 16.08 | 66.91 ± 14.99 |
| ProkBERT | 61.84 ± 8.89 | 83.41 ± 8.29 | 49.52 ± 5.28 | 63.50 ± 7.72 | 31.50 ± 28.50 | 49.34 ± 4.49 | 86.10 ± 8.33 | 56.82 ± 37.42 | 49.07 ± 29.41 | 82.76 ± 12.95 | 81.29 ± 26.40 | 61.87 ± 23.45 | 84.91 ± 8.16 | 85.55 ± 16.14 | 66.74 ± 14.85 |
| ESM-2 | 62.86 ± 12.75 | 83.30 ± 9.29 | 51.30 ± 9.54 | 64.60 ± 8.18 | 31.77 ± 29.34 | 49.38 ± 4.58 | 81.70 ± 11.20 | 56.97 ± 37.48 | 52.70 ± 29.63 | 87.97 ± 10.47 | 89.36 ± 14.86 | **71.41 ± 18.24** | 83.46 ± 11.48 | 86.63 ± 15.62 | 66.11 ± 16.31 |
| ESM-C | 62.71 ± 11.90 | 83.90 ± 8.16 | 50.82 ± 9.24 | 65.37 ± 8.16 | 32.20 ± 29.54 | 49.27 ± 4.24 | 78.71 ± 15.27 | 53.44 ± 40.05 | 55.72 ± 35.81 | 88.05 ± 9.67 | 87.26 ± 20.11 | 67.68 ± 20.43 | 81.79 ± 12.13 | 85.05 ± 16.95 | 61.05 ± 14.90 |
| ProtBERT | 61.93 ± 13.25 | 82.30 ± 10.18 | 50.78 ± 7.31 | 64.86 ± 8.29 | 32.22 ± 29.14 | 50.30 ± 5.05 | 82.07 ± 11.45 | 54.06 ± 39.32 | 54.87 ± 33.97 | 83.27 ± 13.36 | 86.10 ± 18.70 | 68.60 ± 18.70 | 84.71 ± 8.40 | 86.48 ± 14.80 | 68.69 ± 14.17 |
| gLM2 | 60.39 ± 8.39 | 82.71 ± 9.45 | 47.76 ± 4.82 | 62.78 ± 7.22 | 30.97 ± 28.27 | 48.65 ± 4.06 | 71.55 ± 26.57 | 51.10 ± 41.36 | 45.49 ± 27.39 | 77.44 ± 16.90 | 82.79 ± 21.88 | 62.18 ± 20.77 | 82.26 ± 9.24 | 84.22 ± 16.68 | 64.43 ± 16.91 |
| Bacformer | **67.42 ± 16.45** | **86.32 ± 9.03** | **55.09 ± 11.26** | **66.94 ± 8.94** | **33.94 ± 30.24** | **51.99 ± 6.44** | **86.18 ± 9.85** | **58.25 ± 37.22** | **59.53 ± 33.08** | **89.53 ± 10.01** | **91.13 ± 15.07** | 70.20 ± 19.26 | **88.18 ± 9.31** | **89.19 ± 14.12** | **77.15 ± 15.49** |

Supplementary Table 17: Per-phenotype performance (1/2). Values are mean ± standard deviation across 5 random seeds. Bold highlights the best mean per metric. The results were macro-averaged across classes for each phenotype.

Supplementary Table 18: Per-phenotype performance (2/2). Values are mean ± standard deviation across 5 random seeds. Bold highlights the best mean per metric. The results were macro-averaged across classes for each phenotype.

## D    BROADER IMPACT & LIMITATIONS

**Broader impact**    BacBench provides the first public, multi-task testbed that spans gene-, system- and genome-scale prediction problems over 67k genomes from 17.6k species. By standardising data splits, evaluation code and baseline embeddings, it lowers the entry barrier for machine-learning researchers who lack domain-specific pipelines yet want to work on microbial genomics. In the near term, more reliable essential-gene or antibiotic-resistance predictors could shorten drug-development cycles and inform stewardship policies, while better phenotype-from-genome models will accelerate the search for chassis strains that sequester carbon, degrade waste or synthesise valuable biochemicals. Because the benchmark emphasises cross-species generalization, methods that succeed on BacBench are naturally suited to poorly studied or newly sequenced taxa, helping global health laboratories track emerging pathogens even when only draft assemblies are available. Finally, releasing all data under permissive licences and exposing a HuggingFace hub invites continual community contributions, which should foster an open, comparative culture similar to computer-vision or NLP benchmarks and, in turn, drive rapid, reproducible advances in microbial bio-AI research.

**Limitations**    Despite its breadth, BacBench still samples an uneven slice of bacterial diversity: phenotypic-trait labels cluster heavily around medically important genera, and some antibiotic classes remain sparsely represented, which could bias models towards well-studied lineages and mechanisms. Tasks that matter for ecology and biotechnology—horizontal-gene-transfer detection, host–phage interaction, metabolic-flux prediction or transcriptome conditioning—are absent, so performance on BacBench should not be interpreted as general mastery of bacterial genomics. Moreover, the benchmark inherits experimental noise from upstream databases: STRING DB interaction scores mix heterogeneous evidence; operon annotations are incomplete; and phenotype labels amalgamate disparate growth protocols, introducing label uncertainty that caps achievable accuracy. Finally, computing embeddings for every update is resource-intensive, which may hinder participation from groups without access to multi-GPU servers, although smaller surrogate splits are planned for future releases.

