# OpenReview forum: "BacBench: Evaluating Genomic Language Models for Bacteria"
_ICLR.cc/2026/Conference — ICLR 2026 Conference Withdrawn Submission_

### Official Review · Reviewer_wnRy · 2025-11-01

**Soundness:** 3
**Presentation:** 3
**Contribution:** 3
**Rating:** 2
**Confidence:** 4

**Summary:**

This paper introduces BacBench, a large-scale benchmark on bacterial genomes. The authors gathered a massive collection of datasets for 6important biological tasks, ranging from identifying key genes to predicting a bacterium's lifestyle from its DNA. They then use this new benchmark to systematically test a wide variety of modern "genomic language models" (LMs). Their main findings are that different models are good at different things (e.g., DNA models for species identification, protein models for evolutionary tasks), and that while these models are promising, they all struggle to predict the overall function of a bacterium from its genome sequence alone.

**Strengths:**

1.  The biggest strength of this paper is the sheer effort and thoughtful curation that went into creating BacBench. The field of bacterial AI has been fragmented, with everyone using different datasets and tasks. This paper is a huge service to the community.

2.  The paper provides clear takeaways. They show convincingly that models pre-trained specifically on bacteria tend to work better, and that the choice between a DNA-based or protein-based model depends on the tasks.

**Weaknesses:**

1. Is "Gene Essentiality" the Right Question? The benchmark frames the task of finding important genes as binary classification for each gene individually. A more realistic and insightful task would be to see if the models can identify these groups of essential genes like people did in community detection.

2. The results for strain clustering are strange. The models are almost perfect at telling nearly identical species apart, but get much worse at grouping related species into broader families. This suggests that the models might be "cheating" by memorizing very specific DNA signatures, while failing to learn the deeper, shared evolutionary history that biologists care about. This is a critical weakness that the paper reports but doesn't really dig into.

3. The current language models were designed to read text or short DNA fragments. Trying to understand a whole bacterial genome by feeding it to them in small, overlapping chunks and then averaging the results is a very crude approach.

4. A Key Baseline is Missing: For the most important task of predicting a bacterium's phenotype, the authors compare their complex deep learning models against each other but miss a crucial, simple baseline: a classic "bag-of-genes" model.

5. What is the DEG database version? Does the authors use the latest version?

**Questions:**

See weakness above

---

### Official Review · Reviewer_gHfy · 2025-11-01

**Soundness:** 3
**Presentation:** 3
**Contribution:** 3
**Rating:** 4
**Confidence:** 3

**Summary:**

This paper introduces BacBench, a benchmark suite for evaluating machine learning models on bacterial genomics tasks. The authors compile 11 datasets spanning 6 tasks at gene-, system-, and genome-levels, covering 67k genomes from 17.6k species. They evaluate 10 genomic language models (5 DNA LMs, 3 protein LMs, and 2 bacterial LMs) and find that while models show reasonable performance on gene- and system-level tasks, they struggle significantly on whole-genome phenotype prediction. The authors conclude that new approaches are needed to accurately model entire bacterial genomes.

**Strengths:**

* Addresses a real need: The bacterial genomics community indeed lacks standardized, multi-task benchmarks. Most existing resources focus on single species or single tasks.

* Broad taxonomic coverage: 17.6k species across 67k genomes represents genuine taxonomic diversity, enabling evaluation of cross-species generalization.

**Weaknesses:**

*  Ground truth quality is inadequately addressed. The paper acknowledges data quality issues in the Limitations section, yet provides no quantitative analysis of their impact. The operon dataset covers only 5 strains (Section A.2), PPI scores are binarized at an arbitrary 0.6 threshold without sensitivity analysis (Appendix A.3), and phenotype labels from different experimental protocols are merged without standardization (Appendix A.6). The benchmark would be strengthened by: (i) inter-annotator agreement studies, (ii) ablation studies quantifying label noise impact, and (iii) validation against independent datasets where available.

* Evaluation is computationally constrained and incomplete. All genome-level tasks (AMR, phenotype prediction) use only linear probing on frozen representations (Section 4.5), while fine-tuning is acknowledged as computationally prohibitive (Appendix B: ">200 NVIDIA A100 GPUs"). However, essential gene prediction—the only task with fine-tuning results—shows substantial improvements over linear probing (Supplementary Tables 9 vs 10: ProtBERT AUPRC 52.00→73.02, ESM-C 55.08→71.85). This suggests current genome-level results (Table 3: best AUROC 87.61% for AMR; Supplementary Table 15: 71.34% for phenotypes) likely represent lower bounds rather than true model capabilities. The exclusion of Evo from genome-level tasks due to computational cost (Supplementary Table 7: 10.7M GPU-hours vs 1-3K for others) further limits our understanding of whether model scaling benefits these tasks.

* Scalability limitations invalidate genome-level conclusions. Most models handle only 512-2,048 tokens (Table 1), representing <0.1% of typical 3-6 Mbp bacterial genomes. The tiling-and-averaging approach (Appendix B) cannot capture long-range dependencies, syntenic relationships, or multi-locus epistatic effects. Poor genome-level performance may reflect evaluation methodology limitations rather than inherent model deficiencies. The paper conflates "models achieve low scores under current evaluation" with "models fundamentally cannot perform genome-level reasoning"—only the former is supported by evidence. To substantiate the latter, authors would need to show that models with sufficient context (e.g., Evo's 8K tokens) also fail when given complete genomic sequences.

* Results lack mechanistic interpretation. The paper reports extensive numbers but minimal analysis of why models succeed or fail. Key questions remain unanswered: Why do DNA LMs achieve 98.75% ARI at species level but only 68.35% overall (Table 2)? Why does Bacformer excel at operons (77.59% AUROC, Figure 3) but show only moderate improvement on PPI (79.09%, Figure 4)? What explains massive variance across antibiotics (Supplementary Table 13: 20-25% std)? The Discussion (Section 5) proposes generic solutions ("bigger models," "more data") without mechanistic understanding. The benchmark needs feature attribution studies, systematic error analysis, and task difficulty characterization to guide future model development effectively.

**Questions:**

* Ground truth validation: Can you provide inter-rater agreement for at least a subset of operon/phenotype annotations? Or validate against independent datasets?

* Computational budget: What was total GPU-hour cost for this study? This would help community assess reproducibility.

---

### Official Review · Reviewer_DyfM · 2025-11-03

**Soundness:** 3
**Presentation:** 2
**Contribution:** 3
**Rating:** 2
**Confidence:** 5

**Summary:**

The paper “BacBench: Evaluating Genomic Language Models for Bacteria” introduces BacBench, a large-scale, multi-task benchmark designed to systematically evaluate machine learning models in bacterial genomics. It addresses the lack of comprehensive resources capable of testing how well genomic language models generalize across the bacterial tree of life, where prior datasets were limited to single species or narrow prediction tasks.

BacBench integrates eleven datasets covering six prediction tasks that span three biological scales: gene-level (essential gene prediction), system-level (operon identification and protein–protein interaction prediction), and genome-level (strain clustering, antibiotic resistance prediction, and phenotypic trait prediction). Altogether, the benchmark includes 67,000 genomes, 17,600 bacterial species, and 255 million proteins, making it the most extensive bacterial genomics benchmark to date.

The authors evaluate several types of models—DNA language models (DNA LMs), protein language models (pLMs), and bacterial-specific language models (bLMs)—across these tasks. They analyze how model architecture, input modality, and pretraining data influence performance. Their results show that each modeling approach excels at different biological scales: DNA LMs perform best at gene-level tasks, pLMs capture evolutionary and functional relationships useful for clustering and protein–protein interactions, and bLMs such as Bacformer achieve the strongest results for complex genome-level predictions like antibiotic resistance and phenotype inference. However, the study finds that none of the current models can accurately map from whole genomes to phenotypes, highlighting the need for approaches that capture long-range dependencies and interactions across genes and proteins.

To support reproducibility and community development, BacBench also provides a standardized preprocessing, embedding, and evaluation toolkit. The authors suggest that future models should be trained on large, bacteria-specific corpora, integrate DNA, protein, and RNA modalities, and incorporate biological priors such as operon maps or resistance gene catalogs. Overall, BacBench establishes the first unified framework for benchmarking bacterial genomics models and lays the foundation for developing machine learning methods that can reason over entire bacterial genomes and bridge the gap between genotype and phenotype.

**Strengths:**

First, the idea behind BacBench is genuinely valuable and timely. In bacterial genomics, there has long been a lack of standardized benchmarks to evaluate machine learning models. By proposing BacBench, the authors take an important step toward filling that gap. The effort to organize multiple datasets into a unified framework across gene-, system-, and genome-level tasks shows clear initiative and contributes a much-needed foundation for future research in bacterial representation learning.

Second, the paper demonstrates commendable ambition in dataset scale and task diversity. Compiling over 67,000 genomes and 255 million proteins into a cohesive benchmark is no small feat. The inclusion of multiple biological levels reflects an understanding of the hierarchical nature of bacterial systems. Even if the biological coverage is not exhaustive, the benchmark still establishes a structured, multi-level testing environment that future work can build upon.

Finally, the study adds practical value through standardization and accessibility. The release of a preprocessing and evaluation toolkit makes it easier for others to reproduce results and compare models in a consistent way—something often missing in computational biology. The discussion of future directions, including multi-modal modeling and bacteria-specific pretraining, also shows foresight and awareness of where the field is heading.

**Weaknesses:**

First, its biological scope feels narrow. While BacBench is presented as a broad benchmark, most of its tasks focus on familiar, well-trodden areas like operon identification and antibiotic resistance. It doesn’t really capture the full richness of bacterial biology—things like metabolism, gene transfer, or environmental adaptation are missing. Because of that, the benchmark doesn’t fully test how models might generalize to real, complex bacterial systems.

Second, the experimental setup isn’t very consistent or transparent. The paper compares different kinds of language models—DNA-based, protein-based, and bacteria-specific—but it doesn’t carefully control for key factors like model size, pretraining data, or fine-tuning parameters. That makes it hard to tell whether one model is actually better or just trained differently. Some important details are vague, which also hurts reproducibility.

Finally, the analysis lacks depth and biological grounding. The authors report which models perform best but don’t dig into why—there’s little interpretability or discussion of what biological signals the models are learning. And since all the results are purely computational, with no experimental or case-based validation, it’s hard to judge how biologically meaningful the findings really are.

**Questions:**

see weakness

---

### Official Review · Reviewer_fzQ4 · 2025-11-03

**Soundness:** 2
**Presentation:** 3
**Contribution:** 2
**Rating:** 2
**Confidence:** 5

**Summary:**

This paper introduces BacBench, a comprehensive, large-scale benchmark for evaluating genomic language models on bacterial genomics. Motivated by the fragmented, single-species nature of existing benchmarks, BacBench is designed to test cross-species generalization. It consists of 11 datasets across 6 tasks, spanning 17.6k species and organized into gene-, system-, and genome-level challenges. The authors evaluate several existing DNA, protein, and bacteria-specific LMs to establish baseline performance, concluding that bacterial-specific models perform best and that genome-to-phenotype prediction remains a key unsolved problem.

**Strengths:**

BacBench addresses a meaningful gap in genomic language modeling. Bacterial language modeling is growing in interest, yet the field lacks a standardised benchmark across whole-genomes, and current models have not been extensively benchmarked on these problems.

The contribution of new large-scale, curated datasets for whole-genome analysis and benchmarking is commendable. This dataset spans 17.6k species, which, while not comprehensive of all bacteria, is impressive. The datasets seem derived mostly from publicly available resources, but the authors contribute substantial effort through the refinement and pre-processing at scale.

**Weaknesses:**

The paper presents limited machine-learning-specific novelty. This work focuses on long-context, but does not compare or contrast newly designed architectures, such as mamba or hyena, which are specifically aimed at mitigating or solving long-context problems. Whilst benchmark papers do not necessarily need to contribute a new method, the current experimental validation seems too data-focused and does not reference the architectural difference between language-models.

The experimental findings mainly confirm expected trends, that bacterial language models generally perform better on bacterial datasets, and that long-context modeling remains challenging. These observations do not substantially advance the understanding of LM performance in this setting.

For the Protein-Protein interaction task, the authors extract a large amount of protein-protein pairs (640,000) per strain, and do not specify the biological difference between strains. This suggests that homologous proteins from closely related strains (or species, if strains are extracted from the same species) will be present in both training and testing datasets, which may lead to data leakage.

The model selection substantially limits the strength of conclusions on long-context capabilities. The benchmark omits new architectures specifically designed for long-context, such as HyenaDNA and Caduceus. Furthermore, the original Evo model is used, instead of the newly developed Evo 2, which has 1B, 7B and 40B variants, of which, the 1B variant may help mitigate the computational issues with the original Evo for the AMR and Phenotype evaluations. Furthermore, the Evo 2 variants demonstrate significant improvements over the initial model. The experimental validation of purely DNA based models is weakened as a result of this, as it seems natural that the models used (exempting Evo and ProkBERT), will perform poorly due to the poor context window. Utilising smaller alternative DNA LMs such as HyenaDNA, Caduceus, would allow for further investigation into these results.

The authors only use one model that uses Single-Nucleotide Tokenisation, the most common methodology for tokenising DNA-based data. It would be better to include another smaller scale model using SNT to contextualise Evo's performance and better isolate architectural differences from tokenisation methods.

**Questions:**

The authors state "Moreover, even the DNA LMs with very large context window cannot span entire medium-sized bacterial genomes (Brixi et al., 2025).". Given this limitation, could you explicitly clarify the motivation for whole-genome benchmarking with DNA Language Models?

Have the authors considered evaluating performance for Language Models pre-trained on transcriptomics (RNA) rather than purely DNA? It would be interesting to see whether RNA-based models such as OmniGenome or RNA-FM, can transfer effectively to genomic tasks and how they compare to DNA-based models.

Whilst the authors suggest that the whole-genome context in Bacformer improves performance, the empirical performance gap over other models seems quite modest. It would be beneficial for the authors to discuss the potential reasons, such as architecture, model size, training data scale, etc.?

---

### Author Response · Authors · 2025-12-03

Thank you for the feedback given. We are going to improve our paper according to your detailed suggestions.

---

### Note · Authors · 2025-12-03

I have read and agree with the venue's withdrawal policy on behalf of myself and my co-authors.